# FedLOE: Federated Domain Generalization via Locally Overfit Ensemble

**Ruqi Bai**                                                                                     *bai116@purdue.edu*
*Elmore Family School of Electrical and Computer Engineering*
*Purdue University*

**David I. Inouye**                                                                              *dinouye@purdue.edu*
*Elmore Family School of Electrical and Computer Engineering*
*Purdue University*

**Reviewed on OpenReview:** *https://openreview.net/forum?id=W4T9sK6Gai*

## Abstract

In federated learning (FL), clients typically access data from just one distribution. Ideally, the learned models would generalize to out-of-distribution (OOD) data, i.e., domain generalization (DG). However, centralized DG methods cannot easily be adapted to the domain separation context, and some existing federated DG methods can be brittle in the large-client, domain-separated regime. To address these challenges, we revisit the classic mixture-of-experts (MoE) idea by viewing each client as an expert on its own dataset. From this perspective, simple federated averaging can be seen as a type of iterative MoE, where the amount of local training determines the strength of each expert. In contrast to the standard FL communication-performance trade-off, we theoretically demonstrate in linear cases and empirically validate in deep models that reducing communication frequency can effectively enhance DG performance, surpassing centralized counterparts (e.g., +4.34% on PACS). Building on this, we further propose an additional MoE strategy to combine the client-specific classifier heads using standard DG objectives. Our proposed FedLOE method can be viewed as an intermediate approach between FedAvg and one-time ensembling. It demonstrates both theoretical soundness and empirical effectiveness. Moreover, FedLOE requires fewer communication rounds, highlighting its practical efficiency and scalability.

## 1 Introduction

In federated learning (FL), each client owns a portion of the overall data. For example, in a medical network, each hospital may possess its own set of chest X-rays or tissue sample images, collected using different imaging devices. Similarly, regional datasets capturing economic, crime, or demographic information are often compiled with varying methodologies. As a result, each client's dataset may originate from a distinct domain, leading to what is commonly referred to as *domain separation* (Bai et al., 2024). This setup violates the typical independent and identically distributed (i.i.d.) assumption and presents challenges for model training in FL. Additionally, it raises the question: *Can models trained in this setting generalize to clients from previously unseen domains?* This is the central concern of domain generalization (DG), which we consider in the federated setting where domain separation is pronounced and the number of clients is large. This setting is also known to be challenging empirically: the federated DG benchmarking study of Bai et al. (2024) reports that many existing federated DG baselines can substantially degrade under domain separation, especially as the number of clients grows, highlighting a gap between centralized DG assumptions and realistic federated deployment. These findings motivate our focus on mechanisms that remain effective in the large-client, domain-separated regime and do not require sharing data or multi-domain access on any single client.

A straightforward approach might be to adapt centralized DG objectives to each client. However, many popular DG methods assume access to multiple domains during training (Arjovsky et al., 2019; Shi et al., 2022; Sun and Saenko, 2016; Li et al., 2018b; Sagawa et al., 2020; Ganin et al., 2016; Zhou et al., 2021; Li et al., 2018a), which does not hold in the domain separation setting of FL. Existing methods that address DG in FL either do not scale well with many clients (Zhang et al., 2021; Nguyen et al., 2022) or may compromise privacy by sharing the frequency spectra of images (Liu et al., 2021).

To address these challenges, we revisit the mixture-of-experts perspective, treating each client as an expert on its local domain. Rather than being a drawback, domain-specific data may enable clients to specialize, potentially offering complementary information across the network. The key challenge then becomes: *How can these expert models be effectively combined to support out-of-domain generalization?*

We consider two combination strategies. First, we explore an *implicit* combination through the standard FedAvg algorithm and find that communication frequency plays an important role in DG performance (see Section 3). Surprisingly, in some cases, FedAvg with infrequent communication achieves more stable and improved DG performance compared to centralized ERM (which corresponds to communicating at every iteration). We hypothesize that frequent communication may lead models to overfit to common spurious correlations, while infrequent communication allows them to retain diverse representations. Our theoretical analysis supports this behavior in a simplified linear setup (Section 3).

Second, we propose an *explicit* ensembling step for combining locally trained classifier heads. This step is motivated by observations in the DG literature: while DG methods can be effective in simple linear settings (Arjovsky et al., 2019; Sun and Saenko, 2016), they often underperform on overparameterized deep models (Gulrajani and Lopez-Paz, 2021; Koh et al., 2021). Moreover, recent studies suggest that the deep feature extractor may already encode useful invariances, but the classifier head can emphasize spurious features (Wald et al., 2022; Rosenfeld et al., 2022). Drawing on these insights, we adopt a two-stage training process that overfits client-specific classifiers and then combines them using a standard DG objective. Our contributions are as follows:

1. We explore a mixture-of-experts interpretation of federated DG and observe that FedAvg with reduced communication frequency can lead to improved out-of-domain generalization under domain separation.
2. We provide a theoretical analysis in a simplified linear setting that sheds light on the conditions under which our approach may outperform centralized ERM.
3. We introduce a two-stage algorithm, FedLOE, that first performs local overfitting followed by aggregation using parameter averaging and a DG objective, respectively.
4. We present empirical results on several real-world datasets, providing evidence that our approach performs comparably or favorably to existing baselines and is robust to varying numbers of clients.

## 2 Background

Consider a featurizer $g_\theta(\boldsymbol{x}) : \mathbb{R}^D \to \mathbb{R}^d$ with parameters $\theta$ and a linear classifier head denoted as $h_\psi(\boldsymbol{z}) : \mathbb{R}^d \to \mathbb{R}^m$ with parameters $\psi$, where $\boldsymbol{z} = g_\theta(\boldsymbol{x})$. Let $\ell(\cdot, \cdot)$ and $\mathcal{L}(\theta, \psi) := \mathbb{E}_{p(\boldsymbol{x},y)}[\ell(h_\psi(g_\theta(\boldsymbol{x})), y)]$ be the per-sample and expected loss respectively. Let $[A]$ represent the set of integers up to $A$: $[A] \triangleq \{1, 2, \cdots, A\}$. $C$ denotes the number of clients, and $K$ denotes the number of training domains.

### 2.1 Domain generalization (DG)

In domain generalization, we assume there is a set of $K$ domain-specific training distributions where the $k$-th joint distribution is denoted as $p(\boldsymbol{x}, y | k)$ and the marginal probability of each domain is $p(k)$. Additionally, there are one or more domain-specific unseen test distributions denoted by $p(\boldsymbol{x}, y | \tilde{k})$, where $\tilde{k} > K$. The goal of DG is to perform well on these unseen test distributions, which can be formalized as minimizing the expected loss over the set of *unseen* test distributions, i.e.,

$$\min_{\theta, \psi} \sum_{\tilde{k} > K} p(\tilde{k}) \mathcal{L}_{\tilde{k}}(\theta, \psi), \tag{1}$$

where $\mathcal{L}_{\tilde{k}}(\theta, \psi) \triangleq \mathbb{E}_{p(\boldsymbol{x}, y | \tilde{k})}[\ell(h_\psi(g_\theta(\boldsymbol{x})), y)]$. Clearly, because the test domains are unknown, a practical proxy objective is to use standard empirical risk minimization (ERM) by minimizing the expected loss (i.e., risk) of the training domain distributions:

$$\min_{\theta, \psi} \sum_{k=1}^{K} p(k) \mathcal{L}_k(\theta, \psi), \tag{2}$$

where $\mathcal{L}_k$ are the expected losses of the training distributions. Despite its simplicity, simple ERM has been difficult to beat for the DG task (Gulrajani and Lopez-Paz, 2021), particularly when training deep non-linear models. Several common approaches add a type of DG regularization term to the ERM objective.

Common approaches to DG include representation learning, such as domain-invariant representation learning via kernel methods (Muandet et al., 2013; Ghifary et al., 2016), invariant risk minimization (Arjovsky et al., 2019; Krueger et al., 2021), and domain adversarial neural networks (Sun and Saenko, 2016; Li et al., 2018b). Besides domain-invariant learning, several methods add regularizations to the gradient computations (Shi et al., 2022; Rame et al., 2022). Other approaches include distributionally robust optimization (Sagawa et al., 2020), which optimizes for the worst-case distribution scenario across training domains; and meta-learning (Finn et al., 2017; Li et al., 2018a), which uses a learning-to-learn mechanism to acquire general knowledge by constructing meta-learning tasks that simulate domain shifts.

Most of these methods can be viewed as adding a regularization penalty $r(\theta, \psi)$ with tuning parameter $\lambda$ to the simple ERM objective:

$$\min_{\theta, \psi} \sum_{k=1}^{K} p(k) \mathcal{L}_k(\theta, \psi) + \lambda r(\theta, \psi). \tag{3}$$

We select three representative examples that we will use in our experiments. In the IRMv1 method from Arjovsky et al. (2019), the regularization term encourages gradients to be zero:

$$r_{\text{irm}}(\theta, \psi) \triangleq \sum_k \mathbb{E}_{p_k} \|\nabla_{\theta, \psi} \mathcal{L}_k(\theta, \psi)\|^2.$$

Another popular choice is to align the gradients as in Fish (Shi et al., 2022) by computing the inner product of the domain-specific loss functions:

$$r_{\text{fish}}(\theta, \psi) \triangleq - \sum_{k \neq k'} \mathbb{E}_{p_k, p_{k'}} \langle \nabla_{\theta, \psi} \mathcal{L}_k(\theta, \psi), \nabla_{\theta, \psi} \mathcal{L}_{k'}(\theta, \psi) \rangle.$$

The regularization in REx (Krueger et al., 2021) seeks to reduce the loss variance across domains:

$$r_{\text{REx}}(\theta, \psi) \triangleq \text{Var}\left[\mathcal{L}_1(\theta, \psi), \mathcal{L}_2(\theta, \psi), \ldots, \mathcal{L}_K(\theta, \psi)\right].$$

## 2.2 Federated domain generalization

In the federated DG setup, we assume $C$ clients, each possessing data from only a single training domain—a specific form of client heterogeneity known as domain separation (Bai et al., 2024). Formally, client $c$ will have samples from one domain distribution, i.e., $p(\boldsymbol{x}, y | c) \equiv p(\boldsymbol{x}, y | k)$ for some domain $k$. In our experiments, we assume the number of clients is greater than or equal to the number of training domains ($C \geq K$), specifically where $C$ is fairly large in contrast to $K$, which may only be 4 domains (e.g., in PACS). Importantly, this type of client heterogeneity is distinct from label imbalance, i.e., $\exists k \neq k', p(y|k) \neq p(y|k')$, which is the most common heterogeneity considered in the FL literature. Rather, we assume that the joint distribution could be different, i.e., $\exists k \neq k', p(\boldsymbol{x}, y|k) \neq p(\boldsymbol{x}, y|k')$, but we do not assume any particular type of shift between distributions. This is a more general case of client heterogeneity than class imbalance. Crucially, it is important to explicitly distinguish this federated DG setting from standard FL client heterogeneity. Traditional FL generally focuses on mitigating label skew or covariate shift to optimize a global model for the participating clients, or to personalize models for their specific local distributions (i.e., in-domain

performance). In contrast, federated DG under domain separation explicitly targets out-of-distribution (OOD) generalization. The primary objective is not to maximize performance on the $K$ domains present during training, but to extract robust, invariant representations that transfer to completely unseen target domains. Consequently, the core challenge shifts: rather than merely aggregating local knowledge or aligning client updates to satisfy current clients, the system must prevent models from overfitting to common spurious correlations that would degrade performance in novel environments. Like ERM in the centralized case, the standard FL learning algorithm is the FedAvg algorithm (McMahan et al., 2017), which minimizes the expected loss on each client and then averages the parameters from all clients. Prior empirical studies suggest that this federated DG regime is particularly brittle for many existing approaches. For example, Bai et al. (2024) systematically evaluates federated DG methods under domain separation and reports that several baselines may struggle to maintain robust OOD performance in this setting. This motivates our emphasis on (i) communication-efficient mechanisms that scale to many clients and (ii) procedures that preserve client-specific specialization rather than forcing frequent synchronization.

Using our notation, FedAvg can be compactly formalized as performing a sequence of computational steps denoted by $t \in [T]$ where $T$ is the maximum number of computational epochs. In each computational step, FedAvg updates the model and sometimes *also* synchronizes and averages all parameters across clients: for all $c \in [C]$,

$$\left( \psi_c^{t+\frac{1}{2}}, \theta_c^{t+\frac{1}{2}} \right) = \left( \psi_c^t, \theta_c^t \right) - \alpha \nabla \mathcal{L}_c(\psi_c^t, \theta_c^t).$$

$$\left( \psi_c^{t+1}, \theta_c^{t+1} \right) = \begin{cases} \left( \psi_c^{t+\frac{1}{2}}, \theta_c^{t+\frac{1}{2}} \right) & \text{if } t \notin \mathcal{T}, \\ \sum_{c'} \left[ \left( \psi_{c'}^{t+\frac{1}{2}}, \theta_{c'}^{t+\frac{1}{2}} \right) \right] & \text{otherwise,} \end{cases} \quad (4)$$

where $\alpha$ is the step size, and $\mathcal{T} \subseteq [T]$ denotes a set of synchronization indices. If $\mathcal{T} = [T]$ then the synchronization of the sequences is performed every epoch, which corresponds to using mini-batch SGD with mini-batch size $C$ to solve ERM. If $\mathcal{T} = \{T\}$, then (4) amounts to one-shot averaging. A proper choice of the synchronization set $\mathcal{T}$ allows us to obtain an expressive model while keeping the influence of common spurious features relatively low. $\mathcal{T}$ is task-dependent and can be determined experimentally. Furthermore, the averaging weights $p(c')$ are chosen based on the training size.

Formulated in this way, we notice that FedAvg can be interpreted as a type of local SGD (Stich, 2019), which was originally aimed at improving standard supervised learning (i.e., in-domain accuracy). In Section 3, we show that unlike standard supervised learning tasks—where Stich (2019) show that more frequent communication improves in-domain accuracy—the federated DG accuracy is not monotonic w.r.t. communication frequency.

To the best of our knowledge, only a few works seek to solve federated DG. In particular, Nguyen et al. (2022) proposed FedSR, enabling domain generalization while still respecting the distributed and privacy-preserving nature of the FL context by enforcing an $\ell_2$ norm regularizer and a conditional mutual information regularizer on the representation. Liu et al. (2021) propose FedDG, a federated learning paradigm specifically designed for medical image classification. The proposed method requires sharing the amplitude spectrum of images among local clients, which violates standard privacy protocols. Zhang et al. (2021) applies generative adversarial networks (GANs) (Goodfellow et al., 2020) in the FL context, where it first trains the featurizer and classifier by minimizing the empirical loss, then trains the generator and the discriminator using a GAN-based (Goodfellow et al., 2020) approach. Additionally, FedGMA (Tenison et al., 2023) proposes applying a mask on the gradient on the server side to align updates across domains.

## 3 Unraveling the benefits of expert parameter ensembling for domain generalization

In this section, we provide motivation through a simple experiment showing that frequent communication can lead to out-of-distribution (i.e., DG) overfitting. We then show theoretically that infrequent communication leads to better DG accuracy on a linear structural causal model. The linear setting is chosen for closed-form population solutions that disentangle the mechanism of interest from optimization/finite-sample effects. It

is intended as an explanatory and intuitive toy model; our experiments demonstrate the same qualitative behavior in deep networks.

**Motivating observation: infrequent communication leads to better DG accuracy**   We start with a surprising phenomenon related to the effect of communication frequency on DG accuracy when using FedAvg. To demonstrate this phenomenon, we conducted an experiment on PACS with 20 clients using the same model and a fixed computational budget of 256 local epochs, while varying the number of communication rounds (i.e., how many times model parameters are averaged). We follow the standard leave-one-domain-out DG protocol: we designate one PACS domain as the *target* domain that is completely unseen during training, and train only on the remaining *source* domains. We report two test-time metrics: *in-domain* accuracy is computed on the held-out test splits of the source domains, while *DG* (out-of-domain) accuracy is computed on the held-out test split of the target domain. To form 20 clients in the federated setting, we partition the source-domain training data across 20 clients, such that each client contains samples from a single source domain, and run FedAvg under different communication frequencies with the same total local-epoch budget.

As seen in Figure 1, while in-domain accuracy steadily increases and stays nearly constant, the optimal communication frequency for DG is actually a very small number of communications. In terms of DG performance, FedAvg with 4 communications attains the best performance and stability as seen by the low variance. In contrast, its centralized counterpart, mini-batch SGD with a total of 256 communications, performs poorly. These results indicate that increasing communication frequency in federated learning can potentially hinder DG performance.

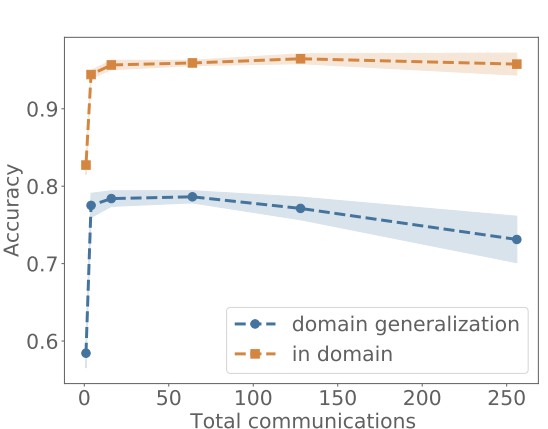

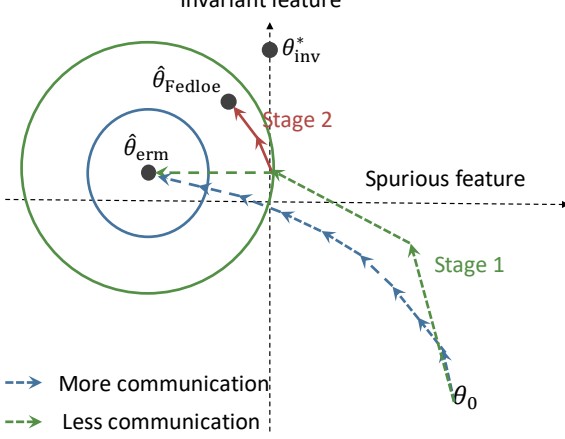

Figure 1: While more communication improves *in-domain* test accuracy (evaluated on held-out test splits of the source/training domains), more communication may harm out-of-distribution domain generalization (*DG*) test accuracy (evaluated on the held-out test split of an unseen target domain). The leftmost points represent one-shot averaging (i.e., only one communication), while the rightmost point corresponds to ERM (i.e., communicate every batch).

Figure 2: This figure illustrates how the infrequent communication of FedLOE's first stage (green) may implicitly regularize the solution away from the ERM solution $\widehat{\theta}_{\text{erm}}$ but closer to the robust invariant solution $\widehat{\theta}_{\text{inv}}$, and FedLOE's second stage (red) explicitly moves toward the invariant solution via DG objectives, while standard FedAvg with frequent communication (blue) converges to the ERM solution.

**Theoretical analysis of expert parameter ensembling for a linear structural equation model** We now theoretically validate our empirical finding by analyzing a linear DG problem to provide further theoretical insight into expert parameter ensembling. Our analysis sheds light on the potential for parameter averaging of locally overfit models (i.e., experts) to indeed improve DG performance. We expect that this result could be extended to more complex linear settings (and possibly non-linear ones) but focus on a simple clear example here. Concretely, we extend the linear model example from Arjovsky et al. (2019) to multiple

invariant and spurious features. Consider the following structural equation model:

$$\boldsymbol{x_{\mathcal{I}}} = \boldsymbol{\xi_1}, \quad y = \boldsymbol{x_{\mathcal{I}}^{\top}}\boldsymbol{\alpha_{\mathcal{I}}} + \xi_2, \quad \boldsymbol{x_{\mathcal{S}}} = y\boldsymbol{\alpha_{\mathcal{S}}} + \boldsymbol{\xi_3}, \tag{5}$$

where $\xi_i, i \in \{1, 2, 3\}$ are exogenous noise variables, $y \in \mathbb{R}$ is the regression target, $\boldsymbol{x_{\mathcal{I}}} \in \mathbb{R}^{|\mathcal{I}|}$ is the invariant feature vector, $\boldsymbol{x_{\mathcal{S}}} \in \mathbb{R}^{|\mathcal{S}|}$ is the spurious feature vector, and $\boldsymbol{\alpha_{\mathcal{I}}} \in \mathbb{R}^{|\mathcal{I}|}$ and $\boldsymbol{\alpha_{\mathcal{S}}} \in \mathbb{R}^{|\mathcal{S}|}$ are the invariant and spurious parameters, respectively. Note that $y$ is causally generated by $\boldsymbol{x_{\mathcal{I}}}$ while $\boldsymbol{x_{\mathcal{S}}}$ are spurious features because they are generated from $y$ (i.e., they are causal descendants).

To frame the DG problem, we assume that each domain or environment shares the above causal model (including the same $\boldsymbol{\alpha_{\mathcal{I}}}$ and $\boldsymbol{\alpha_{\mathcal{S}}}$) but has domain-specific noise distributions for $\boldsymbol{\xi_1}$ and $\xi_2$. Additionally, for the FL context, we assume that each client has access to data from a single domain. We formalize these assumptions next.

**Assumption 3.1** (Finite-moment domain-dependent exogenous noises)**.** We assume the exogenous noises in (5) satisfy:

1. The components of $\boldsymbol{\xi_1}$, $\xi_2$, and $\boldsymbol{\xi_3}$ have finite second moments.
2. $\boldsymbol{\xi_1}$ and $\xi_2$ are zero mean with *domain-dependent* variances:

$$\mathbb{E}[\boldsymbol{\xi_1}] = \boldsymbol{0}, \qquad \mathbb{E}[\boldsymbol{\xi_1}\boldsymbol{\xi_1^{\top}}] = \sigma_1^2(k)\,I, \qquad \mathbb{E}[\xi_2] = 0, \qquad \mathbb{E}[\xi_2^2] = \sigma_2^2(k).$$

3. $\boldsymbol{\xi_3}$ is zero mean with *domain-independent* variance:

$$\mathbb{E}[\boldsymbol{\xi_3}] = \boldsymbol{0}, \qquad \mathbb{E}[\boldsymbol{\xi_3}\boldsymbol{\xi_3^{\top}}] = \sigma_3^2\,I.$$

**Assumption 3.2** (Domain separation)**.** Each client $c \in [C]$ has data from a unique, non-overlapping domain $k \in [K]$ where $C \equiv K$ in this example.

Given this problem context, we consider the OLS estimator of the coefficient $\boldsymbol{\alpha} := [\boldsymbol{\alpha_{\mathcal{I}}}, \boldsymbol{\alpha_{\mathcal{S}}}]^{\top}$ for predicting $y$ from the concatenated features $\boldsymbol{x} = [\boldsymbol{x_{\mathcal{I}}}, \boldsymbol{x_{\mathcal{S}}}]^{\top} \in \mathbb{R}^d$:

$$\widehat{\boldsymbol{\alpha}} \in \arg\min_{\boldsymbol{\alpha} \in \mathbb{R}^d} \mathbb{E}_{(\boldsymbol{x}, y)}\left[(y - \boldsymbol{x}^{\top}\boldsymbol{\alpha})^2\right]. \tag{6}$$

The *domain-invariant*, i.e., robust solution, is to simply ignore the spurious features $\boldsymbol{x_{\mathcal{S}}}$ and only use the linear coefficients of the invariant features, i.e., $\boldsymbol{\alpha}^* := [\boldsymbol{\alpha_{\mathcal{I}}}, \boldsymbol{0}]^{\top}$. This solution also corresponds to finding the true causal mechanism that generates $y$. We compare two distinct cases in this context: (1) the clients communicate every epoch so that the solution converges to the ERM solution (Stich, 2019), or (2) the clients solve their local problems and only communicate once, i.e., one-shot averaging of the model parameters.

**Theorem 3.3.** *Given the least squares problem* (6) *under Assumption 3.1 and Assumption 3.2, one-shot averaging will put relatively more weight on the invariant features, i.e.,*

$$\frac{\|\widehat{\boldsymbol{\alpha}}_{\mathcal{I}}^{\mathrm{ERM}}\|}{\|\widehat{\boldsymbol{\alpha}}_{\mathcal{S}}^{\mathrm{ERM}}\|} < \frac{\|\widehat{\boldsymbol{\alpha}}_{\mathcal{I}}^{\mathrm{OSA}}\|}{\|\widehat{\boldsymbol{\alpha}}_{\mathcal{S}}^{\mathrm{OSA}}\|}, \tag{7}$$

*and OSA will be closer than ERM to the invariant solution* $\boldsymbol{\alpha}^* \triangleq [\boldsymbol{\alpha_{\mathcal{I}}}, \boldsymbol{0}]^{\top}$, *i.e.,*

$$\|\widehat{\boldsymbol{\alpha}}^{\mathrm{ERM}} - \boldsymbol{\alpha}^*\| > \|\widehat{\boldsymbol{\alpha}}^{\mathrm{OSA}} - \boldsymbol{\alpha}^*\|. \tag{8}$$

*Remark* 3.4 (Population-risk setting vs. finite-sample extensions)**.** Theorem 3.3 is stated and proved for the *population* least-squares objective in (6). Accordingly, the proof relies only on the first and second moments in Assumption 3.1 and does not require tail conditions such as sub-Gaussianity. A finite-sample analogue (replacing population moments by empirical estimates on each client/domain) would introduce additional concentration terms depending on the number of clients per domain and the number of samples per client; establishing such bounds typically benefits from stronger distributional assumptions (e.g., sub-Gaussian or bounded-moment conditions), which we leave as future work.

**Corollary 3.5.** *Under the above assumption, the DG risk of ERM is higher than that of OSA. i.e.,*

$$\mathbb{E}\left[(y - \boldsymbol{x}^{\top}\widehat{\boldsymbol{\alpha}}^{\mathrm{ERM}})^2\right] > \mathbb{E}\left[(y - \boldsymbol{x}^{\top}\widehat{\boldsymbol{\alpha}}^{\mathrm{OSA}})^2\right], \tag{9}$$

*where* $(\boldsymbol{x}, y) \sim \mathcal{D}_{\mathrm{test}}$.

While the full proofs are provided in Appendix A and Appendix B, respectively, the key to the proof relies on the fact that the harmonic mean (the ERM solution) is less than the arithmetic mean (the OSA solution). Theorem 3.3 shows that simple one-shot ensembling of local estimators can be more robust than ERM. Furthermore, Appendix B shows that OSA could achieve lower DG risk. This validates our empirical findings and suggests that frequent communication may over-emphasize spurious features. Further theoretical analysis would be required to analyze cases in between these two extremes that are more likely to have better DG performance than either extreme as seen empirically in Figure 1. Given our empirical results, we expect this type of behavior to hold beyond this specific example.

## 4 FedLOE: locally overfit ensemble algorithm for federated DG

In this section, we establish a framework of new algorithms by creating experts via client-specific training followed by expert ensembling via linear parameter combination. Our method aims to reduce overfitting to common spurious features, induce robust features, and control data privacy loss. Our algorithm is designed with two stages that share the same structure: (1) construct expert models by overfitting locally and (2) combine experts via linear combination as outlined in Table 1 and illustrated in Figure 2.

Table 1: FedLOE: An iterative two stage framework for federated DG that locally overfits and ensembles first all model parameters $(\theta, \psi)$ and then only the linear classifier head parameters $\psi$.

|  | Stage 1: Learning shared features | Stage 2: Robustifying classifier head |
|---|---|---|
| Step 1: Locally overfit | SGD on $\theta$ and $\psi$ | SGD on $\psi$ |
| Step 2: Ensemble parameters | Simple average of parameters | Linearly combine via DG objective |

In stage one, we draw from the key observation in Section 3 and reinterpret FedAvg with *infrequent* communication as locally overfitting and then averaging the local expert parameters. Infrequent communication helps to avoid common spurious features while still learning an expressive model. Additionally, infrequent communication is better for communication-constrained FL on edge devices. Even by itself, stage one increases DG robustness compared to synchronizing every mini-batch (which is equivalent to centralized SGD), which requires the most frequent communication. In stage two, we freeze the feature extractor parameters but allow each client to become an expert by training its own linear classifier head locally on its own dataset (note this requires no communication). Then we ensemble the expert classifier heads using standard DG objectives to make the final ensemble classifier more robust.

### 4.1 Stage 1: Learning shared features via FedAvg with infrequent communication

Motivated by the observation in Section 3 that frequent averaging may amplify common spurious features, we use FedAvg with an infrequent communication schedule (4) where $\mathcal{T} = \mathcal{T}_1$. This can be viewed as overfitting each client model on its own data and ensembling these models via parameter averaging where the communication frequency determines how specialized each expert becomes. $\mathcal{T}_1$ is tuned experimentally and generally is a much smaller set than all possible time points. For example, in Figure 1, the optimal choice is $|\mathcal{T}_1| = 4$. By the end of this stage, all clients share a common featurizer $g_{\widehat{\theta}}$ and a linear classifier $\psi^{T_1}$. An ablation study in Section 5.1 also confirms that if we replace stage 1 with the most frequent communications, i.e., mini-batch SGD, the DG accuracy would decrease significantly. This verifies our hypothesis from the exploratory experiments in Figure 1 that frequent communication induces common spurious features. At the end of the first stage, the featurizer parameters are fixed and we will focus on the last linear layer because prior work has shown that the deep featurizer may be robust enough and only the last classifier layer needs to be made robust (Wald et al., 2022).

### 4.2 Stage 2: Robustifying the linear classifier head

In Stage 2, we follow a similar overfitting and ensembling strategy as in Stage 1 but freeze the featurizer and update only the linear classifier head. The key difference is that we compute a specialized linear combination

of the classifier heads via DG objectives rather than a simple average as in FedAvg. To do so, each client randomly holds out a small local *tuning set* (disjoint from its local training data); this tuning set is used only in Stage 2.2 to learn the ensemble weights.

**Stage 2.1: Local overfitting of classifier heads.** In this substage, the clients become experts on their own local datasets by training only their classifier heads $\psi_c, c \in [C]$ on their local datasets $\{(\boldsymbol{x}_i^{\text{train}}, y_i^{\text{train}})\}_{i \in \mathcal{D}_c^{\text{train}}}$. We assume each client $c$ partitions its local labeled data into two disjoint subsets: a *local training set* $\mathcal{D}_c^{\text{train}}$ used in Stage 2.1 to fit $\psi_c$, and a small *local tuning set* $\mathcal{D}_c^{\text{tune}}$ that is held out from training and used only in Stage 2.2 to learn the ensemble weights on the server. Unless otherwise stated, $\mathcal{D}_c^{\text{tune}}$ is formed by a uniform random split of the client's local data and is fixed throughout training for a given run or seed.

While keeping the featurizer $g_{\widehat{\theta}}$ from stage 1 fixed, specifically, for each $c \in [C]$, we initialize at the current classifier head and use SGD to solve

$$\widehat{\psi}_c \triangleq \underset{\psi_c}{\arg\min} \; \mathcal{L}_c(\hat{\theta}, \psi_c) \tag{10}$$

Thus, by the end of substage 2.1, we obtain $C$ distinct linear classifiers $\widehat{\psi}_1, \ldots, \widehat{\psi}_C$, where each $\widehat{\psi}_c \in \mathbb{R}^{m \times d}$.

**Stage 2.2: Robust ensemble of classifier heads via DG methods.** Stage 1 (infrequent-communication FedAvg) is typically the primary driver of DG gains, while Stage 2 provides an *optional*, low-communication refinement of the classifier head. In this stage, we learn a robust linear combination of the expert classifier heads using DG objectives. For instance, IRM (Arjovsky et al., 2019) or Fish (Shi et al., 2022). The additional improvement from Stage 2 can be objective-dependent and dataset-dependent (e.g., whether IRM/Fish/REx are effective in that regime). To do so, the server collects the expert heads $\widehat{\psi}_1, \ldots, \widehat{\psi}_C$ from Stage 2.1 and estimates ensemble weights using a small tuning set from each client. This fine-tuning can be accomplished in two equivalent ways depending on the trust context; additionally, this stage can be omitted if privacy is paramount over performance gains.

**Protocol A (peer-visible heads).** The server broadcasts all expert linear classifiers $\widehat{\psi}_1, \ldots, \widehat{\psi}_C$ to each client. Each client $c$ evaluates every expert $c' \in [C]$ on its local tuning examples $(\boldsymbol{x}, y) \in \mathcal{D}_c^{\text{tune}}$ and sends only the resulting predictions and labels to the server. Concretely, for each client $c$ and each expert $c'$, let $\widehat{y}_{c,c'}(\boldsymbol{x}) \triangleq h_{\widehat{\psi}_{c'}}(g_{\widehat{\theta}}(\boldsymbol{x}))$ denote the logit prediction on input $\boldsymbol{x}$. Each client sends these predictions to the server, and the server then estimates the ensemble weights $\widehat{\phi}$ by minimizing (11).

**Protocol B (trusted server).** Alternatively, to avoid broadcasting client heads to other clients, each client sends only the frozen features $g_{\widehat{\theta}}(\boldsymbol{x})$ and labels for $\boldsymbol{x} \in \mathcal{D}_c^{\text{tune}}$ to the server. The server then computes $\widehat{y}_{c,c'}(\boldsymbol{x})$ locally for all experts $c'$ and solves the same objective (11). Both protocols optimize the same ensemble objective but rely on different trust assumptions.

$$\widehat{\phi} = \underset{\phi \in \mathbb{R}^C}{\arg\min} \; \sum_{c=1}^{C} p(c) \, \mathbb{E}_{(\boldsymbol{x}, y) \sim \mathcal{D}_c^{\text{tune}}} \left[ \ell \left( \sum_{c'=1}^{C} \phi_{c'} \, \widehat{y}_{c,c'}(\boldsymbol{x}), y \right) \right] + \lambda \cdot r(\phi), \tag{11}$$

where $r(\phi)$ denotes a DG regularization term computed from the tuning-set predictions $\{\widehat{y}_{c,c'}(\boldsymbol{x})\}$ and labels across clients (see Appendix C.1 for the specific forms used in our experiments), and $\lambda$ is a regularization parameter.

From another perspective, because both the classifiers and this meta-combination are linear, the resulting ensemble classifier can be simplified to a single linear classifier by taking a combination of the client classifiers, i.e., $\psi_{\text{ensemble}} = \sum_{c=1}^{C} \widehat{\phi}_c \widehat{\psi}_c$. This second perspective allows us to write both steps of this stage in a similar manner to FedAvg with the important exception that the linear combination weights $\widehat{\phi}$ are learned via the DG objective in Equation (11), i.e.,

$$\forall c, \; \widehat{\psi}_c^t = \begin{cases} \widehat{\psi}_c^{t-1} - \eta_c^{t-1} \nabla_\psi \mathcal{L}_c(\hat{\theta}, \widehat{\psi}_c^{t-1}), & \text{if } t \notin \mathcal{T}_2 \\ \sum_{c'=1}^{C} \widehat{\phi}_{c'}^t \widehat{\psi}_{c'}^{t-1}, & \text{if } t \in \mathcal{T}_2. \end{cases} \tag{12}$$

In summary, our FedLOE provides a general framework for domain generalization methods in the federated context, which can incorporate newly designed centralized DG methods in the second stage to provide better DG accuracy. See Section 5 for experiment results. The framework naturally fits into the FL context with a low communication burden. The key idea of the framework is to "free" the clients by letting them overfit in-domain to avoid common spurious features, further personalize their classifiers, and encourage invariant predictions by solving a linear classification problem with DG regularization. See Algorithm 1 in the appendix for more details. Our ablation study in Section 5.1 shows that without Stage 2, or when training Stage 2 with SGD, the DG accuracy would stay the same as FedAvg.

## 5 Experiments

In this section, we evaluate FedLOE on three real-world datasets: PACS (Li et al., 2017), OfficeHome (Venkateswara et al., 2017), and IWildCam-Wilds (Koh et al., 2021). We choose popular IRM (Arjovsky et al., 2019), Fish (Shi et al., 2022), and REx (Krueger et al., 2021) as the DG regularizers for Stage 2, and we compare them against FedAvg (McMahan et al., 2017), FedDG (Liu et al., 2021), FedSR (Nguyen et al., 2022), and FedGMA (Tenison et al., 2023), which were originally designed to solve DG in the FL regime. Our experimental setup follows the domain separation protocol detailed in Bai et al. (2024): each client contains data from a single domain, and we evaluate OOD generalization to an unseen target domain under the standard leave-one-domain-out DG protocol. Note that our focus is federated domain generalization under domain separation, where each client observes data from a single domain and the goal is OOD performance on unseen domains. We therefore compare primarily to methods proposed for federated DG and to FedAvg variants that control communication. We include centralized ERM as contextual upper/lower reference points, but many centralized DG objectives require multi-domain access to compute cross-domain penalties, which is not directly available in the domain separation setting. In Section 5.1, we explore the impact of the number of clients and the effect of Stage 2. For all these experiments, except for the ablation study, we report average performance as well as the standard error over 5 different runs. We explain the most important experimental settings here, but refer the reader to Appendix D for more details. For FedLOE, each client holds out a fixed fraction of its local data as $\mathcal{D}_c^{\text{tune}}$, which is disjoint from $\mathcal{D}_c^{\text{train}}$ for Stage 2.2; all reported in-domain and DG results are computed on the standard held-out test splits.

Table 2: PACS: We show that our methods outperform centralized SGD. Our methods, as well as FedAvg with infrequent communication, are more stable.

| Test Domain | DG accuracy (by domain) | | | | |
| --- | --- | --- | --- | --- | --- |
| | Photo | Art | Cartoon | Sketch | Average |
| Mini-batch SGD | $93.37 \pm 0.52$ | $81.98 \pm 1.71$ | $77.72 \pm 0.67$ | $75.98 \pm 2.92$ | 82.26 |
| FedDG | $96.67 \pm 0.28$ | $84.40 \pm 0.92$ | $75.77 \pm 0.85$ | $74.97 \pm 1.74$ | 82.95 |
| FedSR | $92.63 \pm 0.81$ | $79.73 \pm 1.19$ | $73.90 \pm 3.08$ | $69.93 \pm 0.10$ | 79.05 |
| FedGMA | $97.82 \pm 0.75$ | $88.17 \pm 1.01$ | $77.40 \pm 0.94$ | $79.30 \pm 0.32$ | 85.67 |
| **Our FedAvg** | $97.70 \pm 0.00$ | $\mathbf{88.98 \pm 0.00}$ | $78.63 \pm 0.01$ | $78.61 \pm 0.02$ | 85.98 |
| **FedLOE-IRM** | $97.67 \pm 0.77$ | $86.04 \pm 0.72$ | $\mathbf{80.01 \pm 1.01}$ | $79.01 \pm 0.98$ | 85.68 |
| **FedLOE-Fish** | $\mathbf{98.25 \pm 0.25}$ | $88.62 \pm 0.59$ | $78.53 \pm 0.72$ | $\mathbf{81.01 \pm 0.69}$ | **86.60** |
| **FedLOE-REx** | $97.75 \pm 0.94$ | $86.66 \pm 0.71$ | $78.87 \pm 0.55$ | $79.44 \pm 0.70$ | 85.68 |

### 5.1 Main results

We present the experiment results of our methods as well as the baseline methods on PACS in Table 2, OfficeHome in Table 3, and IWildCam in Table 4. We observe that FedAvg outperforms centralized ERM on all three datasets. This behavior is consistent with prior observations that domain separation is a difficult regime for existing federated DG baselines (Bai et al., 2024), and it highlights the importance of mechanisms such as controlled communication that preserve useful client/domain diversity. This validates our theory that frequent communication across different domains leads to overfitting to spurious features, and our FedLOE method with both stages is comparable to or better than FedAvg on both the PACS and OfficeHome datasets.

Table 3: OfficeHome: We show that our methods outperform centralized SGD. Our methods, as well as FedAvg with infrequent communication, are more stable.

| Test Domain | DG accuracy (by domain) | | | | |
|---|---|---|---|---|---|
| | Art | Clipart | Product | Real-world | Average |
| Mini-batch SGD | $60.06 \pm 2.01$ | $50.32 \pm 1.25$ | $73.20 \pm 1.49$ | $75.63 \pm 1.85$ | 64.80 |
| FedDG | $58.58 \pm 0.74$ | $50.92 \pm 0.89$ | $75.41 \pm 0.43$ | $75.97 \pm 0.98$ | 65.22 |
| FedSR | $1.54 \pm 0.01$ | $1.53 \pm 0.01$ | $1.52 \pm 0.01$ | $1.53 \pm 0.01$ | 1.53 |
| FedGMA | $58.19 \pm 2.74$ | $42.33 \pm 3.65$ | $68.90 \pm 1.64$ | $71.70 \pm 2.07$ | 60.28 |
| **Our FedAvg** | $63.70 \pm 0.01$ | $50.71 \pm 0.02$ | $76.74 \pm 0.00$ | $78.74 \pm 0.01$ | 67.47 |
| **FedLOE-IRM** | $\mathbf{65.55 \pm 0.83}$ | $\mathbf{51.23 \pm 0.45}$ | $76.10 \pm 0.94$ | $\mathbf{78.95 \pm 1.03}$ | **67.95** |
| **FedLOE-Fish** | $64.79 \pm 0.59$ | $50.79 \pm 0.39$ | $\mathbf{76.93 \pm 0.44}$ | $78.4 \pm 0.51$ | 67.72 |
| **FedLOE-REx** | $64.01 \pm 0.29$ | $50.52 \pm 0.38$ | $77.32 \pm 0.34$ | $78.10 \pm 0.42$ | 67.49 |

This suggests that in some cases, our second stage of robust ensembling can improve performance. Furthermore, we achieve the highest performance on the leaderboard of the WILDS Benchmark using ResNet-50 without targeted augmentation in this client heterogeneity setting using infrequent-communication FedAvg. It is also worth noting that our FedLOE-IRM, FedLOE-Fish, and FedLOE-REx methods exhibit less effectiveness compared to FedAvg. This could be attributed to the fact that IRM, Fish, and REx did not demonstrate significant effectiveness in the centralized setting, as reflected in the WILDS benchmark leaderboard (Koh et al., 2021).

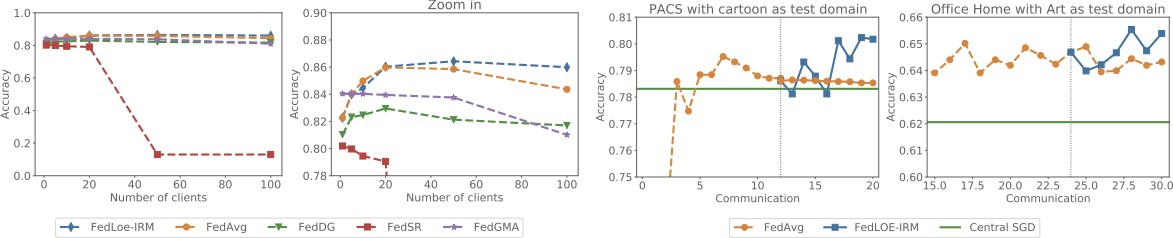

Figure 3: OOD accuracy with a changing number of clients $C$. The second panel is a zoomed-in view of the first figure, showing the details of each method.

Figure 4: DG accuracy per communication on the PACS and OfficeHome datasets; the second stage of our method starts at 12 and 24 respectively (blue).

**The Influence of the Number of Clients $C$:** In this study, we investigate the performance of various algorithms in the FL context as the number of clients $C$ increases on PACS. As shown in Figure 3, OOD accuracy is plotted against the number of clients, where each client only contains samples from a single domain. The results indicate that: **1)** the performance of FedSR degrades as the number of clients increases. **2)** In contrast, our proposed method, FedLOE, not only achieves higher accuracy but also exhibits greater robustness to larger client number settings. We emphasize that our client-scaling plot in Figure 3 is intended to highlight this observed brittleness in a representative baseline (FedSR); we do not claim that all prior federated DG methods necessarily degrade with increasing $C$ under every implementation and tuning.

Table 4: IWildCam: Our methods outperform centralized SGD. Our FedAvg with infrequent communication performs better and is more stable.

| Methods | F1 score |
|---|---|
| Mini-batch SGD | $31.61 \pm 1.88$ |
| FedDG | $31.15 \pm 1.72$ |
| FedSR | $0.01 \pm 0.00$ |
| FedGMA | $26.05 \pm 2.24$ |
| **Our FedAvg** | $\mathbf{33.02 \pm 0.61}$ |
| **FedLOE-IRM** | $31.02 \pm 0.65$ |
| **FedLOE-Fish** | $31.14 \pm 0.91$ |
| **FedLOE-REx** | $31.96 \pm 0.76$ |

**The Effect of Stage 2:** In Figure 4, the DG accuracy per communication is plotted, illustrating the performance boost over FedAvg on both the PACS and OfficeHome datasets. Specifically, the results demonstrate that FedAvg achieves the same accuracy with the same number of communications and computations as our FedLOE-IRM, indicating that FedAvg converges.

## 6 Related work

In this section, we discuss some related approaches.

### 6.1 Federated domain generalization

Federated domain generalization (Federated DG) has emerged as a natural extension of centralized domain generalization, especially suited for real-world applications where data is inherently distributed across multiple domains. In such settings, the need for robust generalization to unseen domains is both intuitive and essential. However, Federated DG introduces two key challenges. First, the presence of non-iid (heterogeneous) data distributions across clients complicates model convergence, even before considering discrepancies between training and testing domains. Second, strict privacy requirements and communication limitations often preclude direct data sharing across domains, thereby rendering many traditional, centralized domain generalization approaches infeasible in federated settings. While standard federated learning algorithms have made significant strides in handling client heterogeneity, their primary objectives fundamentally differ from those of federated domain generalization. Well-established methods such as FedProx (Li et al., 2020) and SCAFFOLD (Karimireddy et al., 2020) are specifically designed to improve optimization stability, handle client drift, and ensure convergence when local data distributions diverge (e.g., under label skew). However, these methods optimize primarily for in-domain performance across the participating training clients; they do not directly encode domain generalization objectives or explicitly encourage representation invariance across domains. Consequently, they are not direct competitors for evaluating out-of-distribution generalization to completely unseen target domains. For this reason, our empirical evaluations focus primarily on methods explicitly designed to tackle the federated DG setting, such as FedDG (Liu et al., 2021), FedSR (Nguyen et al., 2022), and FedGMA (Tenison et al., 2023), which actively attempt to bridge the domain gap under privacy constraints. We view standard heterogeneous FL optimization techniques as complementary to our goals, as they could potentially be integrated alongside DG objectives to further stabilize local training.

### 6.2 Model ensemble

Model ensemble methods aim to improve generalization by combining predictions from multiple models, often trained under varying conditions or on different subsets of data. In the context of Federated DG, ensemble methods have been leveraged to account for domain shifts without requiring centralized data sharing. For instance, FedEM (Marfoq et al., 2021) ensembles personalized models from different clients to form a more generalizable global predictor. FedCE (Cai et al., 2023) continues this line of work by introducing a contrastive ensembling framework that encourages consistency across local models while maintaining domain-specific diversity. These approaches highlight the strength of ensemble-based strategies in mitigating the effects of domain heterogeneity and enhancing robustness to unseen target domains. However, ensembling can incur increased computational and communication overhead, particularly in resource-constrained federated environments, making scalability a key concern.

### 6.3 Personalization and fine-tuning in FL

Personalization aims to improve *per-client in-domain* performance by combining a shared global model with client-specific adaptation. A common and strong baseline is to run FedAvg and then fine-tune locally on each client, which has been widely observed to perform well in heterogeneous FL. Representative personalization approaches include meta-learning-based methods like Per-FedAvg (Fallah et al., 2020) and shared-representation methods such as FedRep (Collins et al., 2021). Recent theory also helps explain why FedAvg followed by local fine-tuning can learn transferable representations across tasks/clients (Collins et al., 2022). Our setting is complementary: we study *domain generalization* to *unseen domains* under domain separation,

without assuming access to target-domain data for adaptation at test time. Nevertheless, personalization and DG are closely related, and combining FedLOE with post-hoc local fine-tuning is an interesting direction for future work.

### 6.4 Benefits and limits of local steps

The role of multiple local steps in FedAvg/local SGD has been studied extensively from an optimization perspective, including conditions under which local updates help or hurt convergence under heterogeneity (Woodworth et al., 2020; Patel et al., 2024). Beyond optimization, recent works suggest that local updates can improve representation learning and transfer, helping explain the empirical strength of FedAvg with fine-tuning (Bao et al., 2024; Collins et al., 2022). These insights are related but do not directly imply our main observation for federated DG: in our domain-separation DG setting, *OOD accuracy is not monotone in communication frequency* and can peak at an intermediate number of communication rounds (Figure 1). Moreover, our linear SCM analysis explicitly models invariant vs. spurious features and shows how averaging locally overfit experts can downweight spurious directions, yielding improved DG risk relative to centralized ERM, see Theorem 3.3.

## 7  Conclusion and discussion

Our paper introduces a novel mixture-of-experts perspective on FL DG in the domain separation context, where each client is viewed as an expert. Our method, while preserving privacy, still shares model parameters with the server. We discuss privacy and potential leakage risks in the Broader Impact section, and note that standard defenses such as secure aggregation and DP can be incorporated. We could further apply differential privacy (Dwork et al., 2006) to strengthen privacy protections.

The key observation is that locally overfitting and then refitting can actually avoid collusion among experts so that they do not find a common set of spurious features—as noted in several recent works. Given our perspective and these observations, we design a novel federated DG method, FedLOE, that combines two stages. The first stage uses the natural and implicit combination strategy of federated averaging to learn a robust deep featurizer. The second stage focuses on making the linear classifier head robust by iteratively locally overfitting the heads and then optimizing the best linear combination of these heads using standard DG objectives. Given that the second stage employs standard DG objectives as subroutines, it offers flexibility that allows us to integrate any future methods aimed at enhancing DG accuracy. More broadly, this work suggests that client heterogeneity in FL may actually be beneficial for OOD generalization, in contrast to the usual assumption that data heterogeneity impairs FL methods. Finally, while our linear structural equation model provides clear theoretical intuition, and our empirical results demonstrate solid gains, extending this theoretical framework to less restrictive models remains an exciting avenue for future work. Additionally, broadening the empirical comparisons to explore how standard heterogeneous FL optimization techniques (e.g., FedProx, SCAFFOLD) might complement or interact with explicit DG objectives could inspire useful follow-up research in robust federated systems.

## Broader impact

This work studies federated domain generalization under domain separation, aiming to improve out-of-distribution robustness without centralizing raw client data. Potential positive impacts include better generalization in safety-critical and high-stakes deployments (e.g., medical imaging across hospitals or sensing across regions), where distribution shift is common.

**Privacy and security considerations.** Our methods do not require sharing raw examples. However, like most FL approaches, they still involve exchanging model information that may leak sensitive attributes. In Stage 1, clients communicate model updates/parameters as in FedAvg, which can be vulnerable to inference attacks (e.g., gradient inversion or membership inference) in adversarial settings.

Stage 2 introduces additional communication that warrants explicit discussion. In Stage 2.1, clients may send their classifier heads to the server; while these heads are smaller than full models, they may still encode information about local data. In Stage 2.2, our two protocols make different trust assumptions: (i) *Protocol A* broadcasts client heads to peers and collects tuning-set predictions/logits and labels; sharing heads and predictions can increase exposure to inference attacks or reveal domain-specific information. (ii) *Protocol B* assumes a trusted server and avoids broadcasting heads to other clients, but may require clients to send tuning-set features and labels to the server, which may also leak private information. In practice, these risks can be mitigated by combining our approach with standard FL privacy mechanisms, such as secure aggregation, encrypted communication, differential privacy (DP) on communicated updates/predictions/features, or restricting the tuning set size and granularity (e.g., sharing only aggregated statistics rather than per-example logits). However, we do not make any explicit formal claims of privacy in this paper and urge that caution be used for any privacy-sensitive applications.

## Acknowledgement

R.B. and D.I. acknowledge support from NSF (IIS-2212097), ARL (W911NF-2020-221), and ONR (N00014-23-C-1016). Any opinions, findings, and conclusions or recommendations expressed in this material are those of the authors and do not necessarily reflect the views of the sponsor.

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

# Appendix

## A Proof of Theorem 3.3

*Proof.* At the solution $\widehat{\boldsymbol{\alpha}}$, we have

$$
\begin{cases}
\left. \frac{\partial L}{\partial \widetilde{\boldsymbol{\alpha}}_{\mathcal{I}}} \right|_{\widetilde{\boldsymbol{\alpha}}_{\mathcal{I}} = \widehat{\boldsymbol{\alpha}}_{\mathcal{I}}} = 0, \\
\left. \frac{\partial L}{\partial \widetilde{\boldsymbol{\alpha}}_{\mathcal{S}}} \right|_{\widetilde{\boldsymbol{\alpha}}_{\mathcal{S}} = \widehat{\boldsymbol{\alpha}}_{\mathcal{S}}} = 0.
\end{cases}
$$

For all invariant features, we have

$$
\begin{aligned}
\left. \frac{\partial L}{\partial \widetilde{\boldsymbol{\alpha}}_{\mathcal{I}}} \right|_{\widetilde{\boldsymbol{\alpha}}_{\mathcal{I}} = \widehat{\boldsymbol{\alpha}}_{\mathcal{I}}} &= 2\mathbb{E}\left[ \boldsymbol{x}_{\mathcal{I}}(\boldsymbol{x}^{\top}\widehat{\boldsymbol{\alpha}} - y) \right] \\
&= 2\mathbb{E}\left[ \boldsymbol{x}_{\mathcal{I}}(\boldsymbol{x}^{\top}\widehat{\boldsymbol{\alpha}} - \boldsymbol{x}_{\mathcal{I}}^{\top}\boldsymbol{\alpha}_{\mathcal{I}} - \xi_2) \right] \\
&\stackrel{(a)}{=} 2\mathbb{E}\left[ \boldsymbol{x}_{\mathcal{I}}(\boldsymbol{x}^{\top}\widehat{\boldsymbol{\alpha}} - \boldsymbol{x}_{\mathcal{I}}^{\top}\boldsymbol{\alpha}_{\mathcal{I}}) \right] \\
&= 2\mathbb{E}\left[ \boldsymbol{x}_{\mathcal{I}}(\boldsymbol{x}_{\mathcal{I}}^{\top}(\widehat{\boldsymbol{\alpha}}_{\mathcal{I}} - \boldsymbol{\alpha}_{\mathcal{I}}) + \boldsymbol{x}_{\mathcal{S}}^{\top}\widehat{\boldsymbol{\alpha}}_{\mathcal{S}}) \right] = 0,
\end{aligned} \tag{13}
$$

therefore, it holds that

$$
\mathbb{E}\left[ \boldsymbol{x}_{\mathcal{I}}\boldsymbol{x}_{\mathcal{I}}^{\top}(\boldsymbol{\alpha}_{\mathcal{I}} - \widehat{\boldsymbol{\alpha}}_{\mathcal{I}}) \right] = \mathbb{E}\left[ \boldsymbol{x}_{\mathcal{I}}\boldsymbol{x}_{\mathcal{S}}^{\top}\widehat{\boldsymbol{\alpha}}_{\mathcal{S}} \right], \tag{14}
$$

which is equivalent to

$$
\mathbb{E}\left[ \boldsymbol{\xi_1}\boldsymbol{\xi_1}^{\top}(\boldsymbol{\alpha}_{\mathcal{I}} - \widehat{\boldsymbol{\alpha}}_{\mathcal{I}}) \right] = \mathbb{E}\left[ \boldsymbol{\xi_1}\boldsymbol{\xi_1}^{\top}\boldsymbol{\alpha}_{\mathcal{I}}\boldsymbol{\alpha}_{\mathcal{S}}^{\top}\widehat{\boldsymbol{\alpha}}_{\mathcal{S}} \right]. \tag{15}
$$

For all spurious features, we have

$$
\begin{aligned}
\left. \frac{\partial L}{\partial \widetilde{\boldsymbol{\alpha}}_{\mathcal{S}}} \right|_{\widetilde{\boldsymbol{\alpha}}_{\mathcal{S}} = \widehat{\boldsymbol{\alpha}}_{\mathcal{S}}} &= 2\mathbb{E}\left[ \boldsymbol{x}_{\mathcal{S}}(\boldsymbol{x}^{\top}\widehat{\boldsymbol{\alpha}} - y) \right] \\
&= 2\mathbb{E}\left[ \boldsymbol{x}_{\mathcal{S}}(\boldsymbol{x}^{\top}\widehat{\boldsymbol{\alpha}} - \boldsymbol{x}_{\mathcal{I}}^{\top}\boldsymbol{\alpha}_{\mathcal{I}} - \xi_2) \right] \\
&= 2\mathbb{E}\left[ \boldsymbol{x}_{\mathcal{S}}(\boldsymbol{x}_{\mathcal{I}}^{\top}(\widehat{\boldsymbol{\alpha}}_{\mathcal{I}} - \boldsymbol{\alpha}_{\mathcal{I}}) + \boldsymbol{x}_{\mathcal{S}}^{\top}\widehat{\boldsymbol{\alpha}}_{\mathcal{S}}) \right] - 2\mathbb{E}\left[ \boldsymbol{\xi_1}^{\top}\boldsymbol{\alpha}_{\mathcal{I}}\boldsymbol{\alpha}_{\mathcal{S}}\xi_2 + \xi_2^2\boldsymbol{\alpha}_{\mathcal{S}} + \boldsymbol{\xi_3}\xi_2 \right] \\
&\stackrel{(b)}{=} 2\mathbb{E}\left[ \boldsymbol{x}_{\mathcal{S}}(\boldsymbol{x}_{\mathcal{I}}^{\top}(\widehat{\boldsymbol{\alpha}}_{\mathcal{I}} - \boldsymbol{\alpha}_{\mathcal{I}}) + \boldsymbol{x}_{\mathcal{S}}^{\top}\widehat{\boldsymbol{\alpha}}_{\mathcal{S}}) \right] - 2\mathbb{E}\left[ \xi_2^2\boldsymbol{\alpha}_{\mathcal{S}} \right] \\
&= 2\mathbb{E}\left[ (y\boldsymbol{\alpha}_{\mathcal{S}} + \boldsymbol{\xi_3}))(\boldsymbol{\xi_1}^{\top}(\widehat{\boldsymbol{\alpha}}_{\mathcal{I}} - \boldsymbol{\alpha}_{\mathcal{I}}) + (y\boldsymbol{\alpha}_{\mathcal{S}} + \boldsymbol{\xi_3}))^{\top}\widehat{\boldsymbol{\alpha}}_{\mathcal{S}}) \right] - 2\mathbb{E}\left[ \xi_2^2\boldsymbol{\alpha}_{\mathcal{S}} \right] \\
&= 2\mathbb{E}\left[ \boldsymbol{\xi_1}^{\top}\boldsymbol{\alpha}_{\mathcal{I}}\boldsymbol{\alpha}_{\mathcal{S}}\left\{ \boldsymbol{\xi_1}^{\top}(\widehat{\boldsymbol{\alpha}}_{\mathcal{I}} - \boldsymbol{\alpha}_{\mathcal{I}}) \right\} \right] + 2\mathbb{E}\left[ \left[ (\boldsymbol{\xi_1}^{\top}\boldsymbol{\alpha}_{\mathcal{I}} + \xi_2)^2\boldsymbol{\alpha}_{\mathcal{S}}\boldsymbol{\alpha}_{\mathcal{S}}^{\top} + \boldsymbol{\xi_3}\boldsymbol{\xi_3}^{\top} \right] \widehat{\boldsymbol{\alpha}}_{\mathcal{S}} \right\} \right] - 2\mathbb{E}\left[ \xi_2^2\boldsymbol{\alpha}_{\mathcal{S}} \right] \\
&= 0,
\end{aligned}
$$

where (b) follows from the independence of $\boldsymbol{\xi_1}, \xi_2$ and $\boldsymbol{\xi_3}$. Therefore, we have

$$
\mathbb{E}\left[ \boldsymbol{\xi_1}^{\top}\boldsymbol{\alpha}_{\mathcal{I}}\boldsymbol{\alpha}_{\mathcal{S}}\left\{ \boldsymbol{\xi_1}^{\top}(\widehat{\boldsymbol{\alpha}}_{\mathcal{I}} - \boldsymbol{\alpha}_{\mathcal{I}}) \right\} \right] + \mathbb{E}\left[ \left[ (\boldsymbol{\xi_1}^{\top}\boldsymbol{\alpha}_{\mathcal{I}} + \xi_2)^2\boldsymbol{\alpha}_{\mathcal{S}}\boldsymbol{\alpha}_{\mathcal{S}}^{\top} + \boldsymbol{\xi_3}\boldsymbol{\xi_3}^{\top} \right] \widehat{\boldsymbol{\alpha}}_{\mathcal{S}} \right\} \right] = \mathbb{E}\left[ \xi_2^2\boldsymbol{\alpha}_{\mathcal{S}} \right]. \tag{16}
$$

Combining (15) and (16), we obtain

$$
\begin{cases}
\mathbb{E}\left[ \boldsymbol{\xi_1}\boldsymbol{\xi_1}^{\top}(\boldsymbol{\alpha}_{\mathcal{I}} - \widehat{\boldsymbol{\alpha}}_{\mathcal{I}} - \boldsymbol{\alpha}_{\mathcal{I}}\boldsymbol{\alpha}_{\mathcal{S}}^{\top}\widehat{\boldsymbol{\alpha}}_{\mathcal{S}}) \right] = 0, \\
\mathbb{E}\left[ \left\{ \boldsymbol{\xi_1}^{\top}\boldsymbol{\alpha}_{\mathcal{I}}\left[ \boldsymbol{\xi_1}^{\top}(\widehat{\boldsymbol{\alpha}}_{\mathcal{I}} - \boldsymbol{\alpha}_{\mathcal{I}}) \right] - \xi_2^2 \right\}\boldsymbol{\alpha}_{\mathcal{S}} \right] + \mathbb{E}\left[ \left[ (\boldsymbol{\xi_1}^{\top}\boldsymbol{\alpha}_{\mathcal{I}} + \xi_2)^2\boldsymbol{\alpha}_{\mathcal{S}}\boldsymbol{\alpha}_{\mathcal{S}}^{\top} + \boldsymbol{\xi_3}\boldsymbol{\xi_3}^{\top} \right] \widehat{\boldsymbol{\alpha}}_{\mathcal{S}} \right] = 0.
\end{cases} \tag{17}
$$

Solving this system yields

$$
\begin{cases}
\boldsymbol{\alpha}_{\mathcal{I}} = \widehat{\boldsymbol{\alpha}}_{\mathcal{I}} + \boldsymbol{\alpha}_{\mathcal{I}}\boldsymbol{\alpha}_{\mathcal{S}}^{\top}\widehat{\boldsymbol{\alpha}}_{\mathcal{S}}, \\
\mathbb{E}\left[ \left\{ (\boldsymbol{\xi_1}^{\top}\boldsymbol{\alpha}_{\mathcal{I}})^2\boldsymbol{\alpha}_{\mathcal{S}}^{\top}\widehat{\boldsymbol{\alpha}}_{\mathcal{S}} + \xi_2^2 \right\}\boldsymbol{\alpha}_{\mathcal{S}} \right] = \mathbb{E}\left[ \left[ (\boldsymbol{\xi_1}^{\top}\boldsymbol{\alpha}_{\mathcal{I}})^2 + \xi_2^2)\boldsymbol{\alpha}_{\mathcal{S}}\boldsymbol{\alpha}_{\mathcal{S}}^{\top} + \boldsymbol{\xi_3}\boldsymbol{\xi_3}^{\top} \right] \widehat{\boldsymbol{\alpha}}_{\mathcal{S}} \right].
\end{cases} \tag{18}
$$

Using the assumption $\mathbb{E}[\xi_i\xi_i^{\top}] = \Sigma_i$,

$$\begin{cases} \boldsymbol{\alpha}_{\mathcal{I}} = \widehat{\boldsymbol{\alpha}}_{\mathcal{I}} + \boldsymbol{\alpha}_{\mathcal{I}} \boldsymbol{\alpha}_{\mathcal{S}}^{\top} \widehat{\boldsymbol{\alpha}}_{\mathcal{S}}, \\ \left( \sigma_2^2 - \boldsymbol{\alpha}_{\mathcal{S}}^{\top} \widehat{\boldsymbol{\alpha}}_{\mathcal{S}} \sigma_2^2 \right) \boldsymbol{\alpha}_{\mathcal{S}} = \Sigma_3 \widehat{\boldsymbol{\alpha}}_{\mathcal{S}}, \end{cases} \tag{19}$$

(19) suggests that our assumption on the noise is mild. As long as either $\xi_2$ or $\boldsymbol{\xi_3}$ depends on the environment, the regressed spurious feature $\widehat{\boldsymbol{\alpha}}_{\mathcal{S}}$ will also depend on the environment. In particular, given that $\boldsymbol{\xi_3}$ is centered i.i.d. sub-Gaussian noise, we have

$$\begin{cases} \boldsymbol{\alpha}_{\mathcal{I}} = \widehat{\boldsymbol{\alpha}}_{\mathcal{I}} + \boldsymbol{\alpha}_{\mathcal{I}} \boldsymbol{\alpha}_{\mathcal{S}}^{\top} \widehat{\boldsymbol{\alpha}}_{\mathcal{S}}, \\ \left( \sigma_2^2 - \boldsymbol{\alpha}_{\mathcal{S}}^{\top} \widehat{\boldsymbol{\alpha}}_{\mathcal{S}} \sigma_2^2 \right) \boldsymbol{\alpha}_{\mathcal{S}} = \sigma_3^2 \widehat{\boldsymbol{\alpha}}_{\mathcal{S}}, \end{cases} \tag{20}$$

which implies

$$\boldsymbol{\alpha}_{\mathcal{S}}^{\top} \widehat{\boldsymbol{\alpha}}_{\mathcal{S}} = \frac{\sigma_2^2 \|\boldsymbol{\alpha}_{\mathcal{S}}\|^2}{\sigma_2^2 \|\boldsymbol{\alpha}_{\mathcal{S}}\|^2 + \sigma_3^2}. \tag{21}$$

Combining (21) with (20), we obtain

$$\begin{cases} \widehat{\boldsymbol{\alpha}}_{\mathcal{I}} = \frac{\sigma_3^2}{\sigma_2^2 \|\boldsymbol{\alpha}_{\mathcal{S}}\|^2 + \sigma_3^2} \cdot \boldsymbol{\alpha}_{\mathcal{I}}, \\ \widehat{\boldsymbol{\alpha}}_{\mathcal{S}} = \frac{\sigma_2^2}{\sigma_2^2 \|\boldsymbol{\alpha}_{\mathcal{S}}\|^2 + \sigma_3^2} \cdot \boldsymbol{\alpha}_{\mathcal{S}}, \end{cases} \tag{22}$$

Therefore, we have

$$\frac{\|\widehat{\boldsymbol{\alpha}}_{\mathcal{I}}\|}{\|\widehat{\boldsymbol{\alpha}}_{\mathcal{S}}\|} = \frac{\sigma_3^2}{\sigma_2^2} \cdot \frac{\|\boldsymbol{\alpha}_{\mathcal{I}}\|}{\|\boldsymbol{\alpha}_{\mathcal{S}}\|}.$$

Regressing solely on the invariant correlations $\widehat{y} = \sum_{i \in \mathcal{I}} x_i \widehat{\alpha}_i$ yields

$$\widehat{\boldsymbol{\alpha}} \in \underset{\alpha_i \in \mathbb{R}, i \in \mathcal{I}}{\arg\min} \, \mathbb{E} \left\| y - \boldsymbol{x}_{\mathcal{I}}^{\top} \boldsymbol{\alpha} \right\|^2, \tag{23}$$

$$\begin{aligned} \frac{\partial L}{\partial \boldsymbol{\alpha}} &= \mathbb{E} \left[ \boldsymbol{x}_{\mathcal{I}} (\boldsymbol{x}_{\mathcal{I}}^{\top} \boldsymbol{\alpha} - y) \right] \\ &= \mathbb{E} \left[ \boldsymbol{x}_{\mathcal{I}} (\boldsymbol{x}_{\mathcal{I}}^{\top} \boldsymbol{\alpha} - \boldsymbol{x}_{\mathcal{I}}^{\top} \boldsymbol{\alpha}_{\mathcal{I}} - \xi_2) \right] \\ &\overset{(a)}{=} \mathbb{E} \left[ \boldsymbol{x}_{\mathcal{I}} \boldsymbol{x}_{\mathcal{I}}^{\top} (\boldsymbol{\alpha} - \boldsymbol{\alpha}_{\mathcal{I}}) \right] = 0. \end{aligned} \tag{24}$$

where (a) follows from the independence of $\boldsymbol{\xi_1}$ and $\xi_2$, and the fact that both $\boldsymbol{\xi_1}$ and $\xi_2$ are centered. Therefore, $\widehat{\boldsymbol{\alpha}} = [\boldsymbol{\alpha_{\mathcal{I}}}, \boldsymbol{0}]^{\top}$.

Consider using ERM to solve (6); i.e., by using all the data $(x_i, y_i)_{i \in [dK]}$, we obtain

$$\sigma_2^2 = \mathbb{E}[\xi_2^2] = \mathbb{E}_{\eta} \left\{ \mathbb{E}_{(\boldsymbol{x}, y)} \left[ \xi_2^2(\eta) \,\middle|\, \eta = i \right] \right\} = \frac{1}{K} \sum_{i=1}^{K} \sigma_2^2(i). \tag{25}$$

where we assume $\eta$ satisfies $\mathbb{P}(\eta = i) = \frac{1}{K}$ for all $i \in [K]$. Note that this is a reasonable assumption given that the amount of data from different distributions is the same. If the training data sizes differ, the distribution of $\eta$ can be adjusted accordingly. Similarly, we have

$$\sigma_3^2 = \frac{1}{K} \sum_{i=1}^{K} \sigma_3^2(i). \tag{26}$$

Therefore, for ERM, we obtain

$$\begin{cases} \widehat{\boldsymbol{\alpha}}_{\mathcal{I}}^{\mathrm{ERM}} = \frac{\sum_{i=1}^{K} \sigma_3^2(i)}{\sum_{i=1}^{K} \sigma_2^2(i) \|\boldsymbol{\alpha}_{\mathcal{S}}\|^2 + \sum_{i=1}^{K} \sigma_3^2(i)} \cdot \boldsymbol{\alpha}_{\mathcal{I}}, \\ \widehat{\boldsymbol{\alpha}}_{\mathcal{S}}^{\mathrm{ERM}} = \frac{\sum_{i=1}^{K} \sigma_2^2(i)}{\sum_{i=1}^{K} \sigma_2^2(i) \|\boldsymbol{\alpha}_{\mathcal{S}}\|^2 + \sum_{i=1}^{K} \sigma_3^2(i)} \cdot \boldsymbol{\alpha}_{\mathcal{S}}. \end{cases} \tag{27}$$

If each client locally uses their own domain data to solve (6), and we then perform one-shot averaging on the solutions, we obtain

$$
\begin{cases}
\widehat{\boldsymbol{\alpha}}_{\mathcal{I}}^{\text{OSA}} = \frac{1}{K} \sum\limits_{i=1}^{K} \frac{\sigma_3^2(i)}{\sigma_2^2(i)\|\boldsymbol{\alpha}_{\mathcal{S}}\|^2 + \sigma_3^2(i)} \cdot \boldsymbol{\alpha}_{\mathcal{I}}, \\
\widehat{\boldsymbol{\alpha}}_{\mathcal{S}}^{\text{OSA}} = \frac{1}{K} \sum\limits_{i=1}^{K} \frac{\sigma_2^2(i)}{\sigma_2^2(i)\|\boldsymbol{\alpha}_{\mathcal{S}}\|^2 + \sigma_3^2(i)} \cdot \boldsymbol{\alpha}_{\mathcal{S}},
\end{cases}
\tag{28}
$$

Comparing the solutions of ERM and one-shot averaging, when $\sigma_3(i) = \sigma_3$, we have

$$
\frac{\sum_{i=1}^{K} \sigma_3^2}{\sum_{i=1}^{K} \sigma_2^2(i)\|\boldsymbol{\alpha}_{\mathcal{S}}\|^2 + \sum_{i=1}^{K} \sigma_3^2} = \frac{K}{\sum_{i=1}^{K} (\sigma_2^2(i)\|\boldsymbol{\alpha}_{\mathcal{S}}\|^2/\sigma_3^2 + 1)}
$$

$$
\leq \frac{1}{K} \sum_{i=1}^{K} \frac{1}{(\sigma_2^2(i)\|\boldsymbol{\alpha}_{\mathcal{S}}\|^2/\sigma_3^2 + 1)}
$$

$$
= \frac{1}{K} \sum_{i=1}^{K} \frac{\sigma_3^2}{\sigma_2^2(i)\|\boldsymbol{\alpha}_{\mathcal{S}}\|^2 + \sigma_3^2},
\tag{29}
$$

where the inequality follows from the fact that the harmonic mean is less than or equal to the arithmetic mean. Therefore, it holds that

$$
\widehat{\boldsymbol{\alpha}}_{\mathcal{I}}^{\text{ERM}} \leq \widehat{\boldsymbol{\alpha}}_{\mathcal{I}}^{\text{OSA}}.
\tag{30}
$$

Furthermore, we have

$$
\frac{\sum_{i=1}^{K} \sigma_2^2(i)}{\sum_{i=1}^{K} \sigma_2^2(i)\|\boldsymbol{\alpha}_{\mathcal{S}}\|^2 + \sum_{i=1}^{K} \sigma_3^2} = \left(1 - \frac{\sum_{i=1}^{K} \sigma_3^2}{\sum_{i=1}^{K} \sigma_2^2(i)\|\boldsymbol{\alpha}_{\mathcal{S}}\|^2 + \sum_{i=1}^{K} \sigma_3^2}\right) \cdot \frac{1}{\|\boldsymbol{\alpha}_{\mathcal{S}}\|^2}
$$

$$
\geq \left(1 - \frac{1}{K} \sum_{i=1}^{K} \frac{\sigma_3^2}{\sigma_2^2(i)\|\boldsymbol{\alpha}_{\mathcal{S}}\|^2 + \sigma_3^2}\right) \cdot \frac{1}{\|\boldsymbol{\alpha}_{\mathcal{S}}\|^2}
$$

$$
= \frac{1}{K} \sum_{i=1}^{K} \frac{\sigma_2^2(i)}{\sigma_2^2(i)\|\boldsymbol{\alpha}_{\mathcal{S}}\|^2 + \sigma_3^2},
\tag{31}
$$

which implies

$$
\widehat{\boldsymbol{\alpha}}_{\mathcal{S}}^{\text{ERM}} \geq \widehat{\boldsymbol{\alpha}}_{\mathcal{S}}^{\text{OSA}}.
\tag{32}
$$

Given that $\sigma_2(i)$ for $i \in [K]$ are distinct, we conclude that the above inequality is strict; i.e.,

$$
\widehat{\boldsymbol{\alpha}}_{\mathcal{I}}^{\text{ERM}} < \widehat{\boldsymbol{\alpha}}_{\mathcal{I}}^{\text{OSA}} \quad \text{and} \quad \widehat{\boldsymbol{\alpha}}_{\mathcal{S}}^{\text{ERM}} > \widehat{\boldsymbol{\alpha}}_{\mathcal{S}}^{\text{OSA}}.
\tag{33}
$$

Recall that $\boldsymbol{\alpha}^* := [\boldsymbol{\alpha}_{\mathcal{I}}, \mathbf{0}]^\top$, then (33) implies

$$
\|\widehat{\boldsymbol{\alpha}}_{\mathcal{I}}^{\text{ERM}} - \boldsymbol{\alpha}_{\mathcal{I}}\|^2 > \|\widehat{\boldsymbol{\alpha}}_{\mathcal{I}}^{\text{OSA}} - \boldsymbol{\alpha}_{\mathcal{I}}\|^2 \quad \text{and} \quad \|\widehat{\boldsymbol{\alpha}}_{\mathcal{S}}^{\text{ERM}} - \mathbf{0}_{\mathcal{S}}\|^2 > \|\widehat{\boldsymbol{\alpha}}_{\mathcal{S}}^{\text{OSA}} - \mathbf{0}_{\mathcal{S}}\|^2.
\tag{34}
$$

Therefore, we have

$$
\|\widehat{\boldsymbol{\alpha}}^{\text{ERM}} - \boldsymbol{\alpha}^*\|^2
$$

$$
= \|\widehat{\boldsymbol{\alpha}}_{\mathcal{I}}^{\text{ERM}} - \boldsymbol{\alpha}_{\mathcal{I}}\|^2 + \|\widehat{\boldsymbol{\alpha}}_{\mathcal{S}}^{\text{ERM}} - \mathbf{0}_{\mathcal{S}}\|^2
$$

$$
> \|\widehat{\boldsymbol{\alpha}}_{\mathcal{I}}^{\text{OSA}} - \boldsymbol{\alpha}_{\mathcal{I}}\|^2 + \|\widehat{\boldsymbol{\alpha}}_{\mathcal{S}}^{\text{OSA}} - \mathbf{0}_{\mathcal{S}}\|^2
$$

$$
= \|\widehat{\boldsymbol{\alpha}}^{\text{OSA}} - \boldsymbol{\alpha}^*\|.
\tag{35}
$$

In addition, we have

$$
\sum_{i=1}^{K} \frac{1}{K} \frac{\sigma_2^2(i)}{\sigma_3^2} \geq \sum_{i=1}^{K} \frac{\sigma_2^2(i)}{\sigma_3^2} q_i
$$

where $q_i$ for $i \in [K]$ is defined as

$$q_i \triangleq \frac{A_i}{\frac{1}{K}\sum_{k=1}^{K} A_k}, \qquad \text{where} \qquad A_i \triangleq \frac{1}{\sigma_2^2(i)/\sigma_3^2\|\boldsymbol{\alpha}_{\mathcal{S}}\|^2 + 1}.$$

Therefore, the robust-to-spurious ratio is given by

$$
\begin{aligned}
\frac{\|\widehat{\boldsymbol{\alpha}}_{\mathcal{I}}^{\text{ERM}}\|}{\|\widehat{\boldsymbol{\alpha}}_{\mathcal{S}}^{\text{ERM}}\|} &= \frac{\sigma_3^2}{\frac{1}{K}\sum_{i=1}^{K}\sigma_2^2(i)} \cdot \frac{\|\boldsymbol{\alpha}_{\mathcal{I}}\|}{\|\boldsymbol{\alpha}_{\mathcal{S}}\|} \\
&= \frac{1}{\frac{1}{K}\sum_{i=1}^{K}\frac{\sigma_2^2(i)}{\sigma_3^2}} \cdot \frac{\|\boldsymbol{\alpha}_{\mathcal{I}}\|}{\|\boldsymbol{\alpha}_{\mathcal{S}}\|} \\
&\leq \frac{1}{\sum_{i=1}^{K}\frac{\sigma_2^2(i)}{\sigma_3^2}q_i} \cdot \frac{\|\boldsymbol{\alpha}_{\mathcal{I}}\|}{\|\boldsymbol{\alpha}_{\mathcal{S}}\|} \\
&= \frac{\frac{1}{K}\sum_{i=1}^{K}A_i}{\frac{1}{K}\sum_{i=1}^{K}\frac{\sigma_2^2(i)}{\sigma_3^2}A_i} \cdot \frac{\|\boldsymbol{\alpha}_{\mathcal{I}}\|}{\|\boldsymbol{\alpha}_{\mathcal{S}}\|} \\
&= \frac{1}{\frac{1}{K}\sum_{i=1}^{K}\frac{\sigma_2^2(i)}{\sigma_3^2}\frac{A_i}{\frac{1}{K}\sum_{k=1}^{K}A_k}} \cdot \frac{\|\boldsymbol{\alpha}_{\mathcal{I}}\|}{\|\boldsymbol{\alpha}_{\mathcal{S}}\|} \\
&= \frac{\|\widehat{\boldsymbol{\alpha}}_{\mathcal{I}}^{\text{OSA}}\|}{\|\widehat{\boldsymbol{\alpha}}_{\mathcal{S}}^{\text{OSA}}\|}.
\end{aligned}
\tag{36}
$$

$\square$

# B  Additional proof that OSA will have better DG accuracy than ERM from linear example

**Corollary B.1.** *Under the assumptions of Theorem 3.3, then the DG risk of ERM is higher than for OSA:*

$$\mathbb{E}_{(\boldsymbol{x},y)\sim\mathcal{D}_{test}}\left[(y - \boldsymbol{x}^{\top}\widehat{\boldsymbol{\alpha}}^{ERM})^2\right] > \mathbb{E}_{(\boldsymbol{x},y)\sim\mathcal{D}_{test}}\left[(y - \boldsymbol{x}^{\top}\widehat{\boldsymbol{\alpha}}^{OSA})^2\right]. \tag{37}$$

*Proof.* We first derive the covariances for a single domain:

$$
\begin{aligned}
\Sigma_{\mathcal{I},\mathcal{I}} &= \sigma_1^2 I \tag{38} \\
\Sigma_{\mathcal{S},\mathcal{S}} &= \mathbb{E}[((\boldsymbol{\alpha}_{\mathcal{I}}^T\xi_1 + \xi_2)\boldsymbol{\alpha}_{\mathcal{S}} + \xi_3)((\boldsymbol{\alpha}_{\mathcal{I}}^T\xi_1 + \xi_2)\boldsymbol{\alpha}_{\mathcal{S}} + \xi_3)^T)] \tag{39} \\
&= \boldsymbol{\alpha}_{\mathcal{I}}^T\mathbb{E}[\xi_1\xi_1^T]\boldsymbol{\alpha}_{\mathcal{I}}\boldsymbol{\alpha}_{\mathcal{S}}\boldsymbol{\alpha}_{\mathcal{S}}^T + \mathbb{E}[\xi_2^2]\boldsymbol{\alpha}_{\mathcal{S}}\boldsymbol{\alpha}_{\mathcal{S}}^T + \mathbb{E}[\xi_3\xi_3^T] \tag{40} \\
&= (\|\boldsymbol{\alpha}_{\mathcal{I}}\|_2^2\sigma_1^2 + \sigma_2^2)\boldsymbol{\alpha}_{\mathcal{S}}\boldsymbol{\alpha}_{\mathcal{S}}^T + \sigma_3^2 I \tag{41} \\
\Sigma_{\mathcal{I},\mathcal{S}} &= \mathbb{E}[x_{\mathcal{I}}x_{\mathcal{S}}^T] = \mathbb{E}[\xi_1((\boldsymbol{\alpha}_{\mathcal{I}}^T\xi_1 + \xi_2)\boldsymbol{\alpha}_{\mathcal{S}} + \xi_3)^T] \tag{42} \\
&= \mathbb{E}[\xi_1\xi_1^T\boldsymbol{\alpha}_{\mathcal{I}}\boldsymbol{\alpha}_{\mathcal{S}}^T + \xi_1\xi_2\boldsymbol{\alpha}_{\mathcal{S}}^T + \xi_1\xi_3^T] \tag{43} \\
&= \sigma_1^2\boldsymbol{\alpha}_{\mathcal{I}}\boldsymbol{\alpha}_{\mathcal{S}}^T. \tag{44}
\end{aligned}
$$

We now decompose the test-domain risk where the domain index is not in the training dataset, i.e., $i \notin [C]$, as follows:

$$\mathbb{E}_{(\boldsymbol{x},y)\sim\mathcal{D}_{\text{test}}}\left[(y - \boldsymbol{x}^\top\widehat{\boldsymbol{\alpha}})^2\right]$$
$$= \mathbb{E}_{(\boldsymbol{x},y)\sim\mathcal{D}_{\text{test}}}\left[(\boldsymbol{x}_{\mathcal{I}}^\top\boldsymbol{\alpha}_{\mathcal{I}} + \xi_2(i) - \boldsymbol{x}^\top\widehat{\boldsymbol{\alpha}})^2\right]$$
$$= \mathbb{E}_{(\boldsymbol{x},y)\sim\mathcal{D}_{\text{test}}}\left[(\boldsymbol{x}^\top\boldsymbol{\alpha^*} + \xi_2(i) - \boldsymbol{x}^\top\widehat{\boldsymbol{\alpha}})^2\right]$$
$$= \mathbb{E}_{(\boldsymbol{x},y)\sim\mathcal{D}_{\text{test}}}\left[(\boldsymbol{x}^\top(\boldsymbol{\alpha^*} - \widehat{\boldsymbol{\alpha}}))^2\right] + \sigma_2^2(i)$$
$$= \mathbb{E}_{(\boldsymbol{x},y)\sim\mathcal{D}_{\text{test}}}\left[(\boldsymbol{\alpha^*} - \widehat{\boldsymbol{\alpha}})^\top \boldsymbol{x}\boldsymbol{x}^\top (\boldsymbol{\alpha^*} - \widehat{\boldsymbol{\alpha}})\right] + \sigma_2^2(i)$$
$$= (\boldsymbol{\alpha^*} - \widehat{\boldsymbol{\alpha}})^\top \mathbb{E}_{(\boldsymbol{x},y)\sim\mathcal{D}_{\text{test}}}\left[\boldsymbol{x}\boldsymbol{x}^\top\right] (\boldsymbol{\alpha^*} - \widehat{\boldsymbol{\alpha}}) + \sigma_2^2(i) \tag{45}$$
$$= (\boldsymbol{\alpha^*} - \widehat{\boldsymbol{\alpha}})^\top \Sigma_{\boldsymbol{x},\boldsymbol{x}}(i) (\boldsymbol{\alpha^*} - \widehat{\boldsymbol{\alpha}}) + \sigma_2^2(i) \tag{46}$$
$$= \begin{bmatrix}(1-\hat{w}_{\mathcal{I}})\boldsymbol{\alpha}_{\mathcal{I}} \\ \hat{w}_{\mathcal{S}}\boldsymbol{\alpha}_{\mathcal{S}}\end{bmatrix}^\top \begin{bmatrix}\Sigma_{\mathcal{I},\mathcal{I}}(i) & \Sigma_{\mathcal{I},\mathcal{S}}(i) \\ \Sigma_{\mathcal{S},\mathcal{I}}(i) & \Sigma_{\mathcal{S},\mathcal{S}}(i)\end{bmatrix} \begin{bmatrix}(1-\hat{w}_{\mathcal{I}})\boldsymbol{\alpha}_{\mathcal{I}} \\ \hat{w}_{\mathcal{S}}\boldsymbol{\alpha}_{\mathcal{S}}\end{bmatrix} + \sigma_2^2(i) \tag{47}$$
$$= \begin{bmatrix}(1-\hat{w}_{\mathcal{I}})\boldsymbol{\alpha}_{\mathcal{I}} \\ \hat{w}_{\mathcal{S}}\boldsymbol{\alpha}_{\mathcal{S}}\end{bmatrix}^\top \begin{bmatrix}\sigma_1^2(i)I_{\mathcal{I}} & \sigma_1^2(i)\boldsymbol{\alpha}_{\mathcal{I}}\boldsymbol{\alpha}_{\mathcal{S}}^\top \\ \sigma_1^2(i)\boldsymbol{\alpha}_{\mathcal{S}}\boldsymbol{\alpha}_{\mathcal{I}}^\top & (\|\boldsymbol{\alpha}_{\mathcal{I}}\|^2\sigma_1^2(i)+\sigma_2^2(i))\boldsymbol{\alpha}_{\mathcal{S}}\boldsymbol{\alpha}_{\mathcal{S}}^\top + \sigma_3^2 I_{\mathcal{S}}\end{bmatrix} \begin{bmatrix}(1-\hat{w}_{\mathcal{I}})\boldsymbol{\alpha}_{\mathcal{I}} \\ \hat{w}_{\mathcal{S}}\boldsymbol{\alpha}_{\mathcal{S}}\end{bmatrix} + \sigma_2^2(i) \tag{48}$$
$$= (1-\hat{w}_{\mathcal{I}})^2\|\boldsymbol{\alpha}_{\mathcal{I}}\|_2^2\sigma_1^2(i) + 2\hat{w}_{\mathcal{S}}(1-\hat{w}_{\mathcal{I}})\|\boldsymbol{\alpha}_{\mathcal{I}}\|_2^2\|\boldsymbol{\alpha}_{\mathcal{S}}\|_2^2\sigma_1^2(i)$$
$$+ \hat{w}_{\mathcal{I}}^2((\|\boldsymbol{\alpha}_{\mathcal{I}}\|^2\sigma_1^2(i)+\sigma_2^2(i))\|\boldsymbol{\alpha}_{\mathcal{S}}\|_2^4 + \sigma_3^2\|\boldsymbol{\alpha}_{\mathcal{S}}\|_2^2) + \sigma_2^2(i) \tag{49}$$

Note that the algorithm specific terms are only via $\hat{w}_{\mathcal{I}}$ and $\hat{w}_{\mathcal{S}}$, because all other terms are constant w.r.t. to the learned parameters $\widehat{\alpha}$. Finally, from the previous theorem proof ((33)), we know that

$$0 \le (1 - \hat{w}_{\mathcal{I}}^{\text{OSA}}) < (1 - \hat{w}_{\mathcal{I}}^{\text{ERM}}) \le 1 \tag{50}$$
$$0 \le \hat{w}_{\mathcal{S}}^{\text{OSA}} < \hat{w}_{\mathcal{S}}^{\text{ERM}} \le 1. \tag{51}$$

Combining these facts with Eqn. 49, we can easily arrive at the result:

$$\mathbb{E}_{(\boldsymbol{x},y)\sim\mathcal{D}_{\text{test}}}\left[(y - \boldsymbol{x}^\top\widehat{\boldsymbol{\alpha}}^{\text{OSA}})^2\right] \tag{52}$$
$$= (1-\hat{w}_{\mathcal{I}}^{\text{OSA}})^2\|\boldsymbol{\alpha}_{\mathcal{I}}\|_2^2\sigma_1^2(i) + 2\hat{w}_{\mathcal{S}}^{\text{OSA}}(1-\hat{w}_{\mathcal{I}}^{\text{OSA}})\|\boldsymbol{\alpha}_{\mathcal{I}}\|_2^2\|\boldsymbol{\alpha}_{\mathcal{S}}\|_2^2\sigma_1^2(i)$$
$$+ (\hat{w}_{\mathcal{I}}^{\text{OSA}})^2((\|\boldsymbol{\alpha}_{\mathcal{I}}\|^2\sigma_1^2(i)+\sigma_2^2(i))\|\boldsymbol{\alpha}_{\mathcal{S}}\|_2^4 + \sigma_3^2\|\boldsymbol{\alpha}_{\mathcal{S}}\|_2^2) + \sigma_2^2(i) \tag{53}$$
$$< (1-\hat{w}_{\mathcal{I}}^{\text{ERM}})^2\|\boldsymbol{\alpha}_{\mathcal{I}}\|_2^2\sigma_1^2(i) + 2\hat{w}_{\mathcal{S}}^{\text{ERM}}(1-\hat{w}_{\mathcal{I}}^{\text{ERM}})\|\boldsymbol{\alpha}_{\mathcal{I}}\|_2^2\|\boldsymbol{\alpha}_{\mathcal{S}}\|_2^2\sigma_1^2(i)$$
$$+ (\hat{w}_{\mathcal{I}}^{\text{ERM}})^2((\|\boldsymbol{\alpha}_{\mathcal{I}}\|^2\sigma_1^2(i)+\sigma_2^2(i))\|\boldsymbol{\alpha}_{\mathcal{S}}\|_2^4 + \sigma_3^2\|\boldsymbol{\alpha}_{\mathcal{S}}\|_2^2) + \sigma_2^2(i) \tag{54}$$
$$= \mathbb{E}_{(\boldsymbol{x},y)\sim\mathcal{D}_{\text{test}}}\left[(y - \boldsymbol{x}^\top\widehat{\boldsymbol{\alpha}}^{\text{ERM}})^2\right]. \tag{55}$$

$$\square$$

## C  FedLOE algorithm pseudo-code

We present our training method FedLOE in Algorithm 1.

### C.1  Penalty choices in the FL setting

For the invariant manager training, in every iteration $t \in \mathcal{T}_2$, we recall (11) for convenience:

$$\widehat{\phi}^t = \underset{\phi\in\mathbb{R}^C}{\arg\min} \sum_c p(c)\mathbb{E}_{p(\hat{y}^t,y^t|c)}^{\text{tune}}\left[\ell\left(\sum_{c'=1}^C \phi_{c'}\widehat{y}_{c'}^t, y^t\right)\right] + \lambda \cdot r\left(\sum_{c'=1}^C \phi_{c'}\widehat{y}_{c'}^t, y^t\right),$$

where $\mathbb{E}_{p(\hat{y},y|c)}^{\text{tune}}[\cdot]$ denotes an empirical average over the tuning set $\mathcal{D}_c^{\text{tune}}$ on client $c$, $\lambda$ is the penalty parameter, and $r(\cdot)$ is a penalty term to induce invariance, which can be chosen using different methods as in (3). For

---

**Algorithm 1** FedLOE: Federated DG via Local Overfitting and Ensembling

---

**Input:** Training datasets $\{x_c^{\text{train}}, y_c^{\text{train}}\}_{c=1}^C$ and fine-tuning datasets $\{x_c^{\text{tune}}, y_c^{\text{tune}}\}_{c=1}^C$ on each of the $C$ clients; iterations for stage 1 and 2: $T_1, T_2$; communication indices $\mathcal{T}_1$ and $\mathcal{T}_2$ for stage 1 and 2; local computations $E_1$ and $E_2$.
  *# Stage 1: Few shot Federated Averaging to Learn $\theta$: (4)*
  *# Stage 2: Robustifying the Linear Classifier Head*
  **for** $t \in \{T_1 + 1, \ldots, T_1 + T_2\}$ **do**
     if $t \notin \mathcal{T}_2$, *# Stage 2.1: Locally overfit classifier heads, distribute to all clients*
     {Client} $\forall c, \widehat{\psi}_c^t = \text{Mini-batch SGD}\left(\widehat{\theta}, \widehat{\psi}_c^{t-1}; E_2\right)$

     {Client $\rightarrow$ Server} $\forall c, \widehat{\psi}_c^t$     {Server $\rightarrow$ Client} $\left\{\widehat{\psi}_c^t\right\}_{c=1}^C$

     Else if $t \in \mathcal{T}_2$, *# Stage 2.2: Robust training*
     *# Step 1 Server collect predictions on fine tuning dataset*
     {Client} $\forall c, \left\{\widehat{y}_{c,c'}^t \triangleq g_{\widehat{\theta}}(x_c^{\text{tune}})\left(\widehat{\psi}_{c'}^t\right)^\top\right\}_{c' \in [C]}$     {Client $\rightarrow$ Server} $\forall c, \{\widehat{y}_{c,c'}^t\}_{c' \in [C]}$
     *# Step 2 Train invariant manager.*
     {Server} solves Equation (11) to obtain $\widehat{\phi}^t$     {Server $\rightarrow$ Client} $\widehat{\psi}^t = \sum_c \widehat{\phi}_c^t \widehat{\psi}_c^t$
  **end for**
  **Return:** $(\widehat{\theta}, \widehat{\psi}) = (\widehat{\theta}^{T_1}, \widehat{\psi}^{T_1 + T_2})$

---

example, to encourage $\sum_{c=1}^C \phi_j \widehat{y}_{c,j}$ to be simultaneously optimal for different client domains $c \in [C]$, we choose $r_{\text{dg}}$ as $r_{\text{irmv}}$, which enforces the gradients to be zero (Arjovsky et al., 2019):

$$r_{\text{irm}}(\theta, \psi) \triangleq \sum_c \mathbb{E}_{p_c} \left\| \nabla_{w|w=1.0}\ell\left(\sum_{c=1}^C w\phi_j \widehat{y}_{c,j}, y_c^{\text{tune}}\right) \right\|^2.$$

Alternatively, we can set $r_{\text{dg}}$ to $r_{\text{fish}}$, which aligns the gradients as in Fish (Shi et al., 2022):

$$r_{\text{fish}}(\theta, \psi) \triangleq -\sum_{c \neq c'} \mathbb{E}_{p_c, p_{c'}} \left\langle \nabla_\phi \ell\left(\sum_{c=1}^C \phi_j \widehat{y}_{c,j}, y_c^{\text{tune}}\right), \nabla_\phi \ell\left(\sum_{c=1}^C \phi_j \widehat{y}_{c',j}, y_{c'}^{\text{tune}}\right) \right\rangle.$$

Furthermore, we can set $r_{\text{dg}}$ to $r_{\text{REx}}$, which reduces the loss variance to 0:

$$r_{\text{REx}}(\theta, \psi) \triangleq \sum_c \mathbb{V}_{p_c} \left[ \ell\left(\sum_{j=1}^C \phi_j Y_{c,j}, y_c^{\text{tune}}\right) \right].$$

# D  Additional experimental details

## D.1  Dataset

**PACS** Gulrajani and Lopez-Paz (2021) is an image dataset designed for domain generalization classification, comprising a total of 4 distinct domains: Photo ($1,670$ images), Art Painting ($2,048$ images), Cartoon ($2,344$ images), and Sketch ($3,929$ images). Within each domain, there are 7 categories present. For this dataset, we use each domain as the test domain in each run, and evaluate the average classification accuracy.

**OfficeHome** Gulrajani and Lopez-Paz (2021) is also an image dataset designed for domain generalization classification containing 4 domains namely Art ($2,427$ images), Clipart ($4,365$ images), Product ($4,439$ images), and Real-World ($4,357$ images) images. Within each domain, there are 65 categories present, which makes it a harder setting than PACS. Similar to PACS, for this dataset, we use each domain as the test domain in each run, and evaluate the average classification accuracy.

**IWildCam** Koh et al. (2021) consists of a diverse collection of wild animals captured by 343 camera traps situated across various natural habitats worldwide. This dataset offers a multi-class classification task,

comprising a total of $203,029$ data samples representing 182 distinct animal species. Our primary objective is to achieve high classification accuracy, particularly for rare animal species. Due to this goal, we employ the macro-F1 score as our chosen metric.

### D.2 Data partitioning

For our experiments on the PACS and OfficeHome datasets, we adopt the following evaluation protocol. We select one domain as the test domain for each experiment, while the remaining 3 domains serve as the training domains. First, we allocate 5% of the training dataset for validation and another 5% for in-domain testing.

Subsequently, we divide the remaining 90% of the dataset among the clients, ensuring that each client exclusively contains data from a single domain. For each client, we utilize 90% of its data for training as $(\boldsymbol{x}^{\text{train}}, y^{\text{train}})$, while the remaining 10% is designated for tuning as $(\boldsymbol{x}^{\text{tune}}, y^{\text{tune}})$. This partitioning strategy enables us to train and evaluate our models on distinct domains while maintaining a balanced and controlled experimental setup.

On the IWildCam dataset, we adhere to the official splits method provided by the WILDS benchmark (Koh et al., 2021) for consistency and comparability. Furthermore, we partition the training data among the clients to ensure that each training domain is present in only one client. This deliberate distribution strategy guarantees maximum domain heterogeneity within the client set.

### D.3 Neural network structure

In all our experiments, we use ResNet-50 (He et al., 2016) as our featurizer, excluding the last fully connected layer. The featurizer produces an output of size 2048. Subsequently, each client maintains its own linear classifier. In stage 1 of our approach, the linear classifiers belonging to each client are averaged using the default weights of FedAvg. In stage 2, we advance this by averaging the linear classifiers using learned weights.

### D.4 Model selection, early stopping and other hyperparameters

In this section, we report the hyperparameters used in the experiments. Please refer to Table 5.

Table 5: Hyperparameters for our infrequent FedAvg and our FedLOE

| Dataset | PACS | OfficeHome | IWildCam |
|---|---|---|---|
| Hyperparameter searching | Leave-one-domain-out-cross-validation | | Best performance on held-out validation dataset |
| Stopping criterion | Fix iteration | | Best performance on held-out validation dataset |
| Batch size | 128 | 128 | 128 |
| Number of iterations at stage 1 | 256 | 256 | 5120 |
| Stage 1 total communication | 8 | 8 | 80 |
| Stage 1 optimizer | ADAM | ADAM | ADAM |
| Stage 1 learning rate | 0.00005 | 0.00003 | 0.00005 |
| Stage 2 num of iterations (FedLOE only) | 8 | 8 | 320 |
| Stage 2 total communication (FedLOE only) | 4 | 4 | 5 |
| Stage 2 optimizer (FedLOE only) | SGD | SGD | SGD |
| Stage 2 learning rate (FedLOE only) | 0.05 | 0.05 | 0.05 |
| Stage 2 weight decay (FedLOE only) | 0.0005 | 0.0005 | 0.0005 |
| IRM $\lambda$ (FedLOE-IRM only) | 100 | 100 | 100 |
| Fish meta lr (FedLOE-Fish only) | 0.05 | 0.05 | 0.01 |
| REx $\lambda$ (FedLOE-Rex only) | 100 | 100 | 100 |

## E Code repository for reproducibility

Please see the code here: https://github.com/inouye-lab/fedloe.

