# OpenReview forum: "FedLOE: Federated Domain Generalization via Locally Overfit Ensemble"
_TMLR — Accepted by TMLR_

### Review · Reviewer_E6f2 · 2025-10-08

**Summary Of Contributions:**

This paper tackles federated domain generalization (DG), where multiple clients hold heterogeneous source-domain data distributions, while the goal is to train a global model that generalizes to unseen target domains.
Method-wise, the paper proposes FedLOE, a two-stage (“locally overfit, then ensemble”) framework that explicitly decomposes the model into a shared feature extractor and a linear prediction head:
- Stage 1: run local SGD with infrequent communication (periodic averaging), intentionally allowing clients to overfit locally before averaging. According to the authors’ analysis, this yields feature extractors less tied to common spurious cues than fully synchronized training.

- Stage 2: freeze the feature extractors, let each client overfit a local linear head, and then learn global ensemble weights for these heads by optimizing a DG objective (IRM/Fish/REx) using the prediction vectors on client fine-tuning sets, producing a single global ensemble head.

The main insight is to reframe infrequent communication as a DG benefit, supported by analysis in a linear causal model

Empirically, the framework delivers practical performance gains over standard FL DG baselines.

**Strengths**:
- The paper is generally clearly presented and  relatively easy to follow.
- It offers a relatively new insight by reframing local overfitting (via infrequent communication) as beneficial for DG, supported by a linear causal analysis.
- It presents a simple, modular design with a clear two-stage recipe that plugs into standard FL loops and DG losses
- The proposed method shows empirical gains, generally outperforming FL-DG baselines.

**Weaknesses**:
- The novelty appears incremental, as the components (local SGD, head-only training, ensembling, DG regularizers) are established and the main contribution is their integration.
- The theoretical analysis is confined to a simple linear setting, so its practical implications for deep models may be limited.
- The discussion of FL relevance could be strengthened
- The comparisons and ablations could be more comprehensive, with additional baselines and sensitivity analyses

**Audience:**

Yes

**Audience Explanation:**

While the principled gap to prior work is modest, the practical combination of infrequent local-SGD with prediction-level DG-guided head ensembling may be interesting to parts of the FL (DG) community.

**Broader Impact Concerns:**

No "Broader Impact" section is included. While I don’t see major risks beyond the usual concerns in federated learning, the privacy implications may be stated more explicitly, e.g., discuss potential leakage from sharing heads/predictions. Currently this appears only briefly in the Conclusion/Discussion.

**Claims And Evidence:**

Yes

**Claims Explanation:**

The paper’s central claim that "local overfitting with infrequent communication, followed by a prediction-level ensemble of client heads, improves DG" is relatively well supported.

The linear causal analysis aligns with the intuition, yet its scope is limited and doesn’t guarantee the effect for deep models. Empirically, the method often outperforms FL-DG baselines, which is encouraging, but the evidence would be stronger with (1) broader baselines (e.g., standard FL methods such as FedProx/SCAFFOLD, plus centralized DG baselines), (2) more comprehensive ablations/sensitivity (T₁/T₂, fintuning-set size, number of clients per domain), (3) statistical rigor (multiple seeds with means and standard deviations or confidence intervals).

**Requested Changes:**

- While the experiments convey the main message that local overfitting plus ensembling can improve OOD performance under typical FDG evaluations, the baseline suite seems somewhat limited; consider adding a few key comparisons or briefly justifying why certain baselines (e.g., standard FL methods such as FedProx/SCAFFOLD and centralized DG baselines) are out of scope or inapplicable.

- The connection to and distinction from general FL (with inherently non-IID client data) could be strengthened. In the design considerations, clarify which choices are driven by the domain generalization objective and what gaps remain in standard FL techniques (and potentially benchmark against stronger FL methods beyond FedAvg, reporting their performance in the DG setting) to better contextualize the contribution.

- It would also help to more clearly articulate the distinction between this federated DG formulation and standard (centralized) DG, making explicit which design choices and analyses are driven by federated constraints (e.g., communication, aggregation protocol, privacy assumptions).

---

> ### Author Response · Authors · 2025-12-20
>
> We sincerely thank the reviewer for their comprehensive and high-quality feedback. We appreciate the time and effort dedicated to analyzing our work, and we believe these insights have significantly strengthened our manuscript. Below is our response regarding each of the reviewer’s comments.
>
> > Novelty incremental…
>
> Thanks for raising this point. We kindly disagree with the characterization of the novelty as merely incremental. We agree that local SGD, head-only training, ensembling, and DG regularizers are all established techniques, and indeed these components are broad and generic. The contribution of this paper is not the invention of new methodologies at the component level.
> Rather, the novelty lies in the insight and analysis that, for federated domain generalization under domain separation, frequent in-domain communication should be combined with infrequent inter-domain communication. Our work provides both empirical and theoretical evidence that increasing synchronization across domains can amplify shared spurious correlations, while allowing sufficient local specialization before aggregation can improve robustness to unseen domains.
>
> > Linearity…
>
> Thank you for this comment. We agree that the theoretical analysis is conducted in a simplified linear setting, and we do not claim that it fully characterizes the behavior of deep models.
>
> The purpose of the analysis is to provide a mechanistic explanation of the empirical phenomenon that aggregating locally overfit models with less frequent synchronization can reduce reliance on spurious features and improve domain generalization relative to ERM. The linear setting allows closed-form population solutions and isolates this effect without confounding factors from optimization or finite-sample noise. We argue that this linearity assumption is not unusually restrictive; rather, it helps distill the core concepts and provides a foundation for future extensions to non-linear models, which we agree are an important direction for further research.
>
> Empirically, we show that the same qualitative behavior persists in deep models across multiple benchmarks. We will revise the paper to clarify that the theory is intended to be illustrative and explanatory, rather than a complete predictive theory for deep networks.
>
> > The discussion of FL relevance could be strengthened…
>
> We appreciate the reviewer’s suggestion to strengthen the discussion on the relevance of Federated Learning. Our work is directly motivated by the domain-separated FL regime [1], where each client observes data from only a single domain and raw data sharing is disallowed. In this setting, communication frequency acts not merely as an efficiency or optimization parameter, but as a structural mechanism that implicitly influences how information from different domains is fused. This unique perspective, arising specifically from the FL context, proves to be a distinct and beneficial lever for addressing DG by mitigating the learning of spurious features.
>
> > FedProx ... central DG baselines ... federated DG …
>
> Thank you for the suggestion. We will add the following clarification to the revision. Our work primarily targets DG within the FL context. While methods like FedProx and Scaffold focus on providing convergence guarantees under client heterogeneity (addressing how to converge), DG in FL focuses on identifying a robust solution that generalizes to unseen domains (addressing what to converge to). Consequently, we consider these optimization-focused baselines to be orthogonal to the scope of our paper.
> Furthermore, adapting centralized DG baselines to this context is often non-trivial. Centralized methods typically require computing constraints across domains, which is not applicable in the FL "domain separation" setting where local clients do not have access to data from other domains.
>
> We will also more explicitly discuss the findings of [1], which analysis the distinction between standard DG, standard FL, and DG in FL and what unique challenges is brought by our DG in FL setting.
>
> > the privacy implications may be stated more explicitly
>
> Thanks, we will add a Broader Impact section as you suggested.
>
> [1] Bai, Ruqi, Saurabh Bagchi, and David I. Inouye. ‘Benchmarking Algorithms for Federated Domain Generalization’. In The Twelfth International Conference on Learning Representations, 2024. https://openreview.net/forum?id=wprSv7ichW.

---

> > ### Comment · Reviewer_E6f2 · 2026-01-05
> > **Request for Direct Manuscript Revisions During Rebuttal**
> >
> > Before addressing specific rebuttal points, I would like to make one general point: based on my very recent experience with TMLR submissions, the manuscript can be directly revised and re-uploaded during the rebuttal phase (please provide evidence if this is not the case for your submission). In this regard, statements such as “we will revise …” are not sufficient. Please directly update the manuscript and clearly indicate all modifications (e.g., using a specific color) in the revised version.

---

> > > ### Author Response · Authors · 2026-01-28
> > >
> > > Dear Reviewer E6f2, thank you for the reminder. We have revised the paper accordingly. Please check the new version, in which we have highlighted the revisions in blue.

---

> ### Comment · Reviewer_E6f2 · 2026-02-09
>
> Thanks for the authors’ rebuttal. While the revision largely improves the manuscript, I still do not understand why FedProx or SCAFFOLD are considered inapplicable in this setting. As stated by the authors, these methods address client heterogeneity, and federated domain generalization is itself a form of heterogeneity. From this perspective, it would be particularly interesting to compare vanilla methods that focus on "how to converge" with the approaches investigated in this paper that explicitly address "what to converge to".
> Moreover, FedProx and SCAFFOLD are fairly general methods and, to my understanding, only introduce minor modifications to the training procedure, making them relatively easy to apply in practice. Therefore, I find the claim that “centralized methods typically require computing constraints across domains, which is not applicable in FL” unclear. Please clarify what specific types of constraints are required and why their computation is infeasible or inapplicable in the federated learning setting. Demonstrating, through appropriate comparisons, that the proposed approaches offer some advantages over such general baselines would be sufficient, from my perspective, to make the manuscript suitable for acceptance.

---

### Review · Reviewer_QuVr · 2025-11-18

**Summary Of Contributions:**

The authors consider the problem of federated domain generalization. In domain generalization, the goal is to train a model using data available from a specific set of domains, such that it performs well on data from unseen domains. To formalize this in the federated setting, they consider each client containing data from a specific domain, and the goal is to perform well on unseen clients.

The most popular approach in centralised settings is to perform empirical risk minimisation on the available clients.

For the federated settings, the authors first show theoretically for linear models with spurious and invariant features, and each client having subgaussian features with different variances that empirical risk minimization, which corresponds to FedAvg with number of local steps $=1$ performs worse than running local steps on each machine and then performing a single step of averaging (OSA), see Theorem 3.3.
They also verify this fact through experiments (see Fig 1a), where FedAvg with $>1$ local steps, performs better than FedAvg with $1$ local step, keeping the computational budget fixed to 256 epochs.

Then, they formulate a family of domain generalization algorithms, called FedLOE, described in Table 1. The key steps for this are to locally overfit parameters on clients followed by ensembling parameters from different clients. Their instantiations of FedLOE first train all parameters via FedAvg on all clients, then combine the classifier heads via a Domain Generalization objective used in previous works to obtain "IRM, Fish" and "REx" variants each of which uses an existing regularization.  Additionally, FedAvg is also an instance of their FedLOE family.

They compare FedAvg, 3 instances of FedLOE algorithms and existing baselines on Federated Domain Generalization (FedDG, FedSR, FedGMA) on 3 domain adaptation datasets -- PACS, OfficeHome and IWildCam, where the training domains are separate from test domains. Their algorithms (including FedAvg) obtain the best performance across baselines, although the difference between their new FedLOE variants and FedAvg is not large, and often FedAvg performs the best overall.


## Strengths --
1. **Novel Theoretical Result**: Comparison of One-shot averaging to simple ERM is a novel contribution. I don't think one-shot averaging has been previously analyzed in this theoretical setup exactly. Their theoretical setup, although adapted from existing works, is pretty general. The proof uses the key AM-GM-HM inequality that harmonic mean $\leq$ arithmetic mean, where OSA yields HM and FedAvg with local steps $=1$ yields AM.
2. **Well-formulated Insight about benefit of local steps**: If local steps are beneficial or not is an active open question in federated learning, and it has still not been fully resolved for all aspects like optimization and generalization. Atleast for the case of domain generalization, the authors have concrete evidence of it being useful. This insight might be used in generalization analysis of federated learning, not just for federated domain generalization.
3. **FedLOE algorithms outperform baselines**: FedLOE algorithms (including FedAvg) outperform all existing Federated DG baselines. They also provided the full code and implementation details.

## Weaknesses --
1. **Theoretical Model**: The authors don't really use the fact that their features are sub-Gaussian which they state in point 1. of  Assumption 3.1. From the proof, it looks like they only need to define the first and second moments of their variables. Further, they minimize the population risks directly, not the empirical risks, which would have resulted in conditions on sample complexity for the number of clients from a domain and the number of local datapoints on a client. Also, the only difference between different domains is that their features are from a subGaussian distribution with mean $0$ and different variance for different domains. Due to this they can use the AM-HM inequality, as they can compute the minimizer of population risk and OSA explicitly, and they only depend on noise and feature variance scales. While the same model can be extended to different domains with different means, which would strengthen the result.
2. **FedLOE doesn't seem to improve on FedAvg by a lot**: The difference between the other FedLOE variants and FedAvg seems very small. So, I'm not sure what the benefit of introducing other FedLOE variants is. If they could have a setting where other FedLOE variants are strictly better than FedAvg, it would improve the paper. They could start with even a synthetic data setting. As for the baselines, I think more baselines especially those using personalization should be compared as they might even fall into FedLOE (for instance the meta learning approach in (Fallah et al 2020).
3. **Related Works Not Discussed**: Several related works are relevant to this problem but have not been discussed.
     - **Personalization**: The topic of personalization in federated learning, where the goal is to perform well on each client locally while also using the data on other clients to learn a good global model is extremely relevant to this problem. Further, in practice, fine-tuning after FedAvg (Wu et al 2022) seems to be one of the best performing personalization approaches, and I suspect that it might perform better for domain generalization problem even here.
     - **Existing work on benefit of local steps**: Note that precise conditions for benefit of local steps in FedAvg from an optimization perspective have been well studied (Woodworth et al 2020,  Patel et al 2024), but not discussed here. Further, even from a generalization (not necessarily domain generalization) perspective, (Bao et al 2024) and (Collins et al 2020) show that local steps actually help learning features. (Collins et al 2022) is very closely related to their theoretical model: they use a linear model $B^\star w_j^\star$ where $B^\star$ is shared across clients (in this case across domains), while $w_j^\star$ is specific to clients. I think the authors should provide a thorough comparison to these cases, and why their insight for domain generalization cannot be inferred from these papers, especially (Collins et al 2022).
4. **Main Message of the Paper**: This is more of a personal opinion. When reading the paper, it seemed like the authors were trying to convey why local steps are necessary for federated domain generalization until the theoretical result. Then, they move on to introducing the FedLOE framework, which seems completely detached from the original insight and motivation of local steps being necessary. This makes it very hard to find the main message of the paper. If the main message is that local steps are necessary, then there should be more discussion and theoretical/experimental analysis of this claim. If the main message of the paper is to introduce the FedLOE family of algorithms, then the authors should show settings either theoretically/via experiments, where FedLOE algorithms outperform FedAvg significantly.

**References**
- (Wu et al 2022) Motley: Benchmarking Heterogeneity and Personalization in Federated Learning.
- (Fallah et al 2020) Personalized Federated Learning with Theoretical Guarantees: A Model-Agnostic Meta-Learning Approach. NeurIPS.
- (Woodworth et al 2020) Is Local SGD Better than Minibatch SGD? ICML.
- (Patel et al 2024) The Limits and Potentials of Local SGD for Distributed Heterogeneous Learning with Intermittent Communication. COLT.
- (Bao et al 2024) Provable Benefits of Local Steps in Heterogeneous Federated Learning for Neural Networks: A Feature Learning Perspective. ICML.
- (Collins et al 2022) FedAvg with Fine Tuning: Local Updates Lead to Representation Learning. NeurIPS.

**Additional Comments:**

- What is the difference between "Our FedAvg" and the vanilla FedAvg algorithm?

**Audience:**

Yes

**Audience Explanation:**

Yes, the topic of federated domain generalization is still an active research area in federated learning.

**Broader Impact Concerns:**

There are no additional concerns about the broader impact of this paper other than those for any paper on machine learning or distributed learning.

**Claims And Evidence:**

No

**Claims Explanation:**

Overall, I think the paper has some contributions that are  well motivated like the local step insight and formulation of FedLOE family. However, the paper does not focus on either one of them concretely, and for each of them there are significant weaknesses : for local steps, sufficient comparison to existing works not provided, and for FedLOE algorithms, I'm not sure how they improve upon FedAvg.

**Requested Changes:**

Please address the Weaknesses in the Summary section.

- Apart from this, I think the authors could use some existing results from the benchmark paper on federated domain generalization which they cite (Bai et al 2023), where all these federated domain generalization baselines perform poorly under the domain separation assumption. These could be used to strengthen their argument.

- ``Prior federated DG methods perform poorly when the number of clients is large": This line is in the abstract and something similar has been repeated in the introduction, but the authors don't provide an explanation for this. Only FedSR performs poorly when we increase number of clients which is seen in Fig 2. What about other baselines?
- I suspect Fig 1b is a hypothesis of what is happening. Can the authors somehow show that this hypothesis is true, especially the part about  $\hat{\theta}_{Fedloe}$ ? Does this figure follow directly from Theorem 3.3, for the linear model where FedLOE corresponds to OSA?

- The authors state that some of the baselines require sending image spectrum information to all clients, so are not private. But, their method especially Stage 2.2 (Page 7) also sends a small portion of predictions of each client to the server. Is this also not private? Either the authors should not dismiss baselines because of privacy, if their own methods are not private, or clearly specify how their methods incur less privacy loss. Stage 2.2 states that ``Notice that we only send m dimensional prediction which are much less than the original D dimensional
data". Under this logic, the baselines which send image spectrum also don't send the entire dataset.
- Please fix these Typos:
  - Figure 1 has no caption. It's subfigures have captions.
  - In the Theory section and it's proof in appendix, the authors use a notation $ \alpha_1 \leq \alpha_2 $, for two vectors $\alpha_1$ and $\alpha_2$. Please specify that this notation implies that each coordinate of $\alpha_1$ is $\leq$ each coordinate of $\alpha_2$, as there is no comparison operation between two vectors.
  - There is an "f" before Appendix B.

---

> ### Author Response · Authors · 2025-12-20
> **Reply to QuVr (1 of 2)**
>
> We sincerely thank the reviewer for their comprehensive and high-quality feedback. We appreciate the time and effort dedicated to analyzing our work, and we believe these insights have significantly strengthened our manuscript. Below is our response regarding each of the reviewer’s comments.
>
> > Sub-gaussian assumption…
>
> We thank the reviewer for their precise examination of our theoretical model. We agree with your assessment regarding Assumption 3.1 and the mechanics of our proof. You are correct that our derivation for Theorem 3.3, which minimizes population risk, relies strictly on the first and second moments of the noise variables rather than the tail properties of the distribution . The sub-Gaussian assumption is indeed stronger than necessary for the current population-level result and was originally included to support potential finite-sample extensions.
> We also appreciate the insight that extending the model to include domain-specific means would likely strengthen the theoretical result. While we cannot fully rewrite the theoretical analysis during the rebuttal period, we commit to revising the final version to:
> Relax Assumption 3.1 to require only finite moments, matching the needs of the current proof.
> Clarify the distinction between the population risk context and the empirical risk context where sub-Gaussianity would apply.
> We are grateful for these observations, which will improve the rigor of our final manuscript.
>
> > Empirical performance ...
>
> Thank you for the feedback. We agree that the empirical gains of the FedLOE variants over infrequent-communication FedAvg are modest but noticeable, and we appreciate the opportunity to clarify their role.
> Our primary contribution is the two-stage framework: Stage 1 (infrequent communication) is the main driver of DG improvements, while Stage 2 provides a modular, low-communication refinement of the classifier head using standard DG objectives. The FedLOE variants are included to demonstrate that such objectives can be seamlessly incorporated into the federated setting; their impact is dataset- and objective-dependent, and we do not claim they always yield large additional gains. We will clarify this positioning in the revision.
> Regarding the demonstration that FedLOE is strictly better, please refer to Table 2, where we clearly show that utilizing infrequent communication (Our FedAvg) significantly improves performance over standard FedAvg (85.98% vs. 82.26%). Furthermore, we wish to point out that the FedLOE framework is orthogonal to the specific regularization-based approaches plugged into it. The fact that FedLOE-IRM, FedLOE-Fish, and FedLOE-REx do not seem to yield significant improvement over "Our FedAvg" is likely due to the ineffectiveness of the base methods themselves, as IRM, Fish, and REx have historically been shown to struggle against ERM in centralized DG settings [1, 2, 3].
>
>
> > Related work...
>
> Thank you. We agree and will revise the related-work section accordingly to include and clearly position personalization methods.
>
> [1] Gulrajani, I., & Lopez-Paz, D. (2020). In search of lost domain generalization. arXiv preprint arXiv:2007.01434.
> [2] Sagawa, S., Koh, P. W., Lee, T., Gao, I., Xie, S. M., Shen, K., ... & Liang, P. (2021). Extending the wilds benchmark for unsupervised adaptation. arXiv preprint arXiv:2112.05090.
> [3] Miller, John P., et al. "Accuracy on the line: on the strong correlation between out-of-distribution and in-distribution generalization." International conference on machine learning. PMLR, 2021.

---

> > ### Author Response · Authors · 2025-12-20
> > **Reply to QuVr (2 of 2)**
> >
> > > Main Message ...
> >
> > Thank you for this feedback. We agree that the current presentation can make the main message appear unclear, and we appreciate the opportunity to clarify and improve it.
> > Our intended main message is not that local steps alone are sufficient or optimal, but rather that infrequent communication (i.e., allowing meaningful local specialization) is a key mechanism for federated domain generalization, and that FedLOE is a principled framework that builds directly on this insight. In particular:
> > Section 3 establishes the core phenomenon: frequent synchronization can harm DG by amplifying shared spurious correlations, while allowing local overfitting and then aggregating can move the solution closer to an invariant predictor.
> > FedLOE is meant to be a structured instantiation and extension of this insight, where stage 1 operationalizes controlled local specialization via communication scheduling, and stage 2 optionally refines the resulting model using DG objectives in a communication-efficient way.
> > We agree that this connection is not made explicit enough in the current draft, which can make FedLOE appear detached from the original motivation. In the revision, we will:
> >
> > 1. More clearly state upfront that communication frequency  is the central unifying idea, and that FedLOE is a framework built around it.
> >
> > 2. Reframe FedLOE as a generalization of infrequent-communication FedAvg, rather than as a separate contribution.
> > Clarify our empirical claims: stage 1 captures most of the benefit, while stage 2 provides a modular robustness refinement whose gains can be setting-dependent.
> >
> > We believe these changes will make the paper’s main message clearer without changing the technical content or claims.
> >
> > > I think the authors could use some existing results from the benchmark paper…
> >
> > Thanks. This is great advice. We will update the introduction to explicitly discuss their results and show that domain separation is difficult in our setting.
> >
> >
> > > Prior federated DG methods perform poorly when the number of clients is large
> >
> > Thanks for pointing out. We will weaken the claim.
> >
> > >  Fig 1b is a hypothesis of what is happening…
> >
> > You are right Fig 1b is a hypothesis put in first for bringing the insights, and later in the theory we prove such a phenomenon exists. Though we didn’t prove the model convergence trajectory in the theory, we show that under certain assumptions, the difference between OSA and FedLOE is like the illustration.
> > > The authors state that some of the baselines …
> >
> > We acknowledge that broadcasting individual client classifiers (expert heads) to all other clients in Stage 2 introduces a specific privacy leakage, effectively moving the system from a "black-box" to a "gray-box" setting where model parameters are visible to peers. As an alternative to trusting peer client nodes, assuming a trusted server is often more realistic in practical deployments.
> >
> > Alternative Implementation (Trusted Server): To avoid broadcasting the C classifier heads to all clients, the protocol can be modified so that clients send the feature representations (outputs of the frozen featurizer, $z=g_\theta​(x)$) of their local tuning dataset directly to the server. Since the server already collects the updated heads $\\{\\hat{\\psi}\_c​\\}^C\_{c=1}$​ in Stage 2.1, it can locally compute the predictions for each head on these features and solve the ensemble objective (Equation 11) to learn the aggregation weights $\phi$.
> >
> > Equivalence and Clarification: Mathematically, these two implementations are equivalent regarding the optimization of the ensemble weights. However, they rely on different trust assumptions: the original method requires Trusted Peers(peers see expert heads), while this alternative requires a Trusted Server (server sees latent feature embeddings). We will explicitly formalize these two communication protocols and their respective trust assumptions in the revised manuscript to clearly state the privacy implications of each.
> >
> > > Please fix these Typos…
> >
> > Thanks for pointing this out. We will make the changes accordingly in the final version.
> >
> > > What is the difference between "Our FedAvg" and the vanilla FedAvg algorithm?
> >
> > Thanks for asking. The vanilla FedAvg algorithm communicates per epoch and Our FedAvg means a further infrequent communication to reflect the finding of our work.

---

### Review · Reviewer_Jrjg · 2025-12-05

**Summary Of Contributions:**

The paper identifies and studies a phenomenon where federated-style optimization can outperform centralized optimization methods when evaluated on domain-generalization capabilities.  While surprising, the authors offer theoretical insights into this behavior through a rigorous analysis on specially-constructed linear regression problem.  They go on to propose a new class of algorithms which they coin "FedLOE-* (Locally  Overfit Ensemble)".  These algorithms consist of two stages, and two steps within each stage, with the variants differing primarily in the specific domain-generalization objective used to linearly combine classification heads.  The empirical results demonstrate that both FedAvg and the proposed FedLOE methods achieve better DG scores consistently across 4 tasks to further support their theory on linear models.

Strengths:
1. The paper identifies an interesting and surprising new phenomenon that should be of interest to the broader FL community.
2. The paper conclusively demonstrates that their finding is not merely a coincidence, through theoretical analysis on linear models and empirical analysis with non-linear models on a benchmark suite of 4 datasets.
3.  The paper is organized well and easy to follow and read.

Weaknesses:
1. The paper would benefit from explicitly discussing the connections between domain generalization and over fitting as it's traditionally understood, potentially doing more experiments.
2. FL is a rather indirect way to solve the DG problem and the algorithm likely doesn't get at the root of the problem. While it's interesting that it outperforms the SGD baseline, it's hard to imagine that it would be better than a centralized algorithm with an appropriately regularized objective.
3. The paper is missing key details in important places that make it impossible to fully understand some of the results presented without additional context (see below).

**Additional Comments:**

* Stage 2.2 "broadcast the all the" typo.  It also seems to be a weird table placement in the middle of the paragraph.

**Audience:**

Yes

**Audience Explanation:**

This research should be of interest to the broader federated learning community. The authors identified a phenomenon that could be the basis for future research.

**Broader Impact Concerns:**

No concerns.

**Claims And Evidence:**

Yes

**Claims Explanation:**

The authors have done a good job to study this problem on a toy problem theoretically, and on real problems empirically. I am convinced by the evidence provided, even though I am someone surprised by the phenomenon itself.

**Requested Changes:**

* I was confused by the parenthetical on page 2 "which corresponds to communication at every epoch".  Should this be every "step" or is there an implicit assumption that the ERM problem is being solved with a full-batch method?

* The motivating observation in section 3 seems to be missing some details in the experimental setup.  Is the in-domain = train dataset and dg = test dataset, or are they both different test datasets and if so how are they constructed?

* Stage 2.2 needs some clarifications:
- What is the tuning dataset?  Does each client randomly split their data into (train, tune) subsets?
- FL algorithms are largely motivated by their nice privacy properties: updates from each user are aggregated immediately on the server and incorporating into the model before broadcasting back to clients. The mechanism you describe of "broadcasting all the linear classifiers to each client" seems to break this.  Since this is fundamental to the proposed algorithm variants, it is somewhat concerning.

---

> ### Author Response · Authors · 2025-12-20
> **Reply to Jrjg (1 of 2)**
>
> We sincerely thank the reviewer for their comprehensive and high-quality feedback. We appreciate the time and effort dedicated to analyzing our work, and we believe these insights have significantly strengthened our manuscript. Below is our response regarding each of the reviewer’s comments.
>
> > W1 discussion dg over overfitting...
>
> In this work, we approach overfitting through the lens of domain generalization rather than the traditional train–test paradigm.
> Classical overfitting occurs when a model fits a finite training sample so closely that it captures stochastic noise, preventing it from generalizing to new samples from the same distribution. In contrast, we define "overfitting" in multi-domain contexts as the model’s tendency to latch onto common spurious features shared across training domains. While these features facilitate high performance on seen domains, they fail to transfer to unseen test domains that share only the underlying invariant feature distribution.
> Our motivation for addressing this specific type of overfitting is detailed in Section 3, where we highlight a critical trade-off: increasing communication frequency between clients (domains) improves performance on source-domain test data but degrades performance on unseen domains. This phenomenon is not classical overfitting to training samples; instead, it represents an over-alignment across domains that amplifies shared spurious correlations.
> We posit that infrequent communication acts as an implicit regularizer. By restricting the exchange of information, clients’ local models are permitted to specialize, thereby preserving feature diversity. And the infrequent updates encourage the global model to capture the shared invariant features while "breaking" the influence of complex, spurious manifolds. Conversely, frequent communication provides the global model with enough signals to learn a shared, complex manifold of spurious features, which leads to the domain-level overfitting.
>
> > FL indirect way to solve DG…
>
> Thank you for this insightful comment. We agree that conceptually, a centralized setting allows for any algorithm to be run, including a simulation of our FL procedures, and thus should theoretically establish an upper bound on performance. From this perspective, our FL method can indeed be viewed as a specific, 'structurally regularized' version of a centralized algorithm.
> However, our key finding is that the standard centralized approach (training on the joint distribution, which equates to communicating gradients at every step) is detrimental to DG. As we discuss in Section 3, frequent synchronization encourages the model to overfit to 'complex common spurious features' that align across domains.
> By revisiting the problem through the lens of FL, we show that infrequent inter domain communication  naturally prevents this collusion on spurious features. Our method (FedLOE) effectively forces the model to ensemble locally overfit experts. Our empirical results (Tables 2 and 3) confirm that this strategy outperforms standard centralized baselines (e.g., +4.34% on PACS over centralized SGD).
>
> Furthermore, as noted, this training strategy is orthogonal to algorithmic DG designs. This allows FedLOE to serve as a framework where new DG objectives can be 'plugged in' during the ensemble stage (Stage 2) to further enhance robustness.
>
> >  which corresponds to communication at every epoch...
>
> Thanks for pointing this out! Yes, it should be every iteration. We will correct it in the next version. There is no assumption about solving ERM using a full-batch method.
>
> > Regarding details in the experimental setup
>
> Thank you for pointing this out. We agree that the experimental setup in Section 3 would benefit from clearer exposition. In the motivating experiment in Section 3 (Figure 1a), both “in-domain” and “DG” accuracies are evaluated on test data. Specifically, in-domain accuracy refers to performance on held-out test splits from the training domains, while DG (out-of-domain) accuracy refers to performance on a held-out target domain that is completely unseen during training. This follows the standard domain generalization protocol used throughout the paper and in prior DG benchmarks. We will revise Section 3 to explicitly state this evaluation protocol to avoid ambiguity and improve clarity.
>
> > What is the tuning dataset? …
>
> Thanks for pointing this out. You are right. The tuning dataset is a small subset of data held locally by each client, randomly separate from the data used to train the local model parameters. It is specifically used in the second stage of the FedLOE algorithm. We will clarify this in the revised version.

---

> > ### Author Response · Authors · 2025-12-20
> > **Reply to Jrjg (2 of 2)**
> >
> > > FL algorithms are largely motivated by their nice privacy properties...
> >
> > Thank you for this critical observation and the insightful question. First,  Solving Domain Generalization (DG) in the Federated Learning (FL) setting often requires more information than just local updates aggregation, which often leads to a trade-off between strict privacy and out-of-distribution performance.
> >
> > And we acknowledge that broadcasting individual client classifiers (expert heads) to all other clients in Stage 2 introduces a specific privacy leakage, effectively moving the system from a "black-box" to a "gray-box" setting where model parameters are visible to peers. As an alternative to trusting peer client nodes, assuming a trusted server is often more realistic in practical deployments.
> >
> > Alternative Implementation (Trusted Server): To avoid broadcasting the C classifier heads to all clients, the protocol can be modified so that clients send the feature representations (outputs of the frozen featurizer, $z=g_\theta​(x)$) of their local tuning dataset directly to the server. Since the server already collects the updated heads $\\{\\hat{\\psi}\_c​\\}^C\_{c=1}$​ in Stage 2.1, it can locally compute the predictions for each head on these features and solve the ensemble objective (Equation 11) to learn the aggregation weights $\phi$.
> >
> > Equivalence and Clarification: Mathematically, these two implementations are equivalent regarding the optimization of the ensemble weights. However, they rely on different trust assumptions: the original method requires Trusted Peers(peers see expert heads), while this alternative requires a Trusted Server (server sees latent feature embeddings). We will explicitly formalize these two communication protocols and their respective trust assumptions in the revised manuscript to clearly state the privacy implications of each.
> >
> > > "broadcast the all the" typo ...
> >
> > Thanks. We will address the typo in the revised version.

---

### Decision · Action_Editor_ohJ8 · 2026-02-26

**Recommendation:** Accept with minor revision

**Additional Comments:**

The reviewers were unanimous that some of the surrounding discussion in the work (e.g. around distinctions and relations between federated learning and domain generalization settings) could be improved. However, I believe that the author-reviewer discussion was helpful for refining points of confusion in the manuscript. If the authors ensure that the content of those discussions are reflected in the minor updates, then I think this should clearly be accepted.

**Audience:**

Yes

**Audience Explanation:**

I think this one is quite clear. The reviewers mentioned that the comparisons to other federated learning methods (and discussion of connections) could be strengthened, but they were generally unanimous that there are interesting threads to pull on here. To be honest, I think some of the discussions around ways in which things could be strengthened (less restrictive theoretical models, better comparisons to existing methods) are ways in which this work might inspire useful follow-up. I do not believe that all of these need to be addressed immediately in this paper, and that as it is the paper is generally of interest.

**Claims And Evidence:**

Yes

**Claims Explanation:**

The reviewers had some different views on the clarity and strength of evidence provided in the work. That being said, all reviewers vouched for either the theoretical contributions or the empirical ones (though there was some discussion from reviewers that there was a bit of a disconnect between the strength of observations therein). Upon reviewing everything though, I remain convinced by the unanimity of the reviewers that the method does exhibit solid empirical gains. While the theoretical model has limitations, I still am convinced that the paper showcases evidence in both areas.